# Circulating ACE2-expressing extracellular vesicles block broad strains of SARS-CoV-2

Lamiaa El-Shennawy[1,19], Andrew D. Hoffmann[1,19], Nurmaa Khund Dashzeveg[1,19], Kathleen M. McAndrews[2,19], Paul J. Mehl[3], Daphne Cornish[1], Zihao Yu[1], Valerie L. Tokars[1], Vlad Nicolaescu[4], Anastasia Tomatsidou[4], Chengsheng Mao[5], Christopher J. Felicelli[6], Chia-Feng Tsai[7], Carolina Ostiguin[3], Yuzhi Jia[1], Lin Li[8], Kevin Furlong[4], Jan Wysocki[9], Xin Luo[2], Carolina F. Ruivo[2], Daniel Batlle[9], Thomas J. Hope[10], Yang Shen[11], Young Kwang Chae[12], Hui Zhang[8], Valerie S. LeBleu[1,2,13], Tujin Shi[7], Suchitra Swaminathan[3,14], Yuan Luo[5], Dominique Missiakas[4], Glenn C. Randall[4], Alexis R. Demonbreun[1], Michael G. Ison[15,16], Raghu Kalluri[2,17,18,20✉], Deyu Fang[3,6,20✉] & Huiping Liu[1,3,12,20✉]

The severe acute respiratory syndrome coronavirus 2 (SARS-CoV-2) has caused the pandemic of the coronavirus induced disease 2019 (COVID-19) with evolving variants of concern. It remains urgent to identify novel approaches against broad strains of SARS-CoV-2, which infect host cells via the entry receptor angiotensin-converting enzyme 2 (ACE2). Herein, we report an increase in circulating extracellular vesicles (EVs) that express ACE2 (evACE2) in plasma of COVID-19 patients, which levels are associated with severe pathogenesis. Importantly, evACE2 isolated from human plasma or cells neutralizes SARS-CoV-2 infection by competing with cellular ACE2. Compared to vesicle-free recombinant human ACE2 (rhACE2), evACE2 shows a 135-fold higher potency in blocking the binding of the viral spike protein RBD, and a 60- to 80-fold higher efficacy in preventing infections by both pseudotyped and authentic SARS-CoV-2. Consistently, evACE2 protects the hACE2 transgenic mice from SARS-CoV-2-induced lung injury and mortality. Furthermore, evACE2 inhibits the infection of SARS-CoV-2 variants (α, β, and δ) with equal or higher potency than for the wildtype strain, supporting a broad-spectrum antiviral mechanism of evACE2 for therapeutic development to block the infection of existing and future coronaviruses that use the ACE2 receptor.

A full list of author affiliations appears at the end of the paper.

Despite the tremendous success of the COVID-19 vaccine development, the pandemic caused by SARS-CoV-2 has been challenging due to fast-evolving mutant strains[1–5] and slow vaccination globally. Shortly after the outbreak, over 1000 SARS-CoV-2 variants were detected[6] and several mutant strains (α, β, and δ) dominated with higher infection rates and/or fatality than the wild-type (WT) strain[4,5,7,8]. Fully vaccinated populations only reached below 30% worldwide and about half in the US as of September 2021. Moreover, the risk of future emerging coronaviruses infecting human always exist. To better protect vulnerable people, both unvaccinated and vaccinated, it is urgent to develop novel therapeutics that can broadly target distinct strains of evolving SARS-CoV-2 and future coronaviruses.

Similar to other coronaviruses such as SARS-CoV, which caused an outbreak in 2003[9], the WT and mutant strains of SARS-CoV-2 infect host cells such as human pneumocytes via the entry receptor angiotensin-converting enzyme 2 (ACE2)[1,10–12]. The mutations-caused alterations in the viral proteins such as the attachment protein—spike glycoprotein (S), in particular the external receptor-binding domain (RBD) in the variants (α, β, and δ), render a greater binding affinity than the WT in binding to ACE2[1,4,5,10–12]. Approaches to block or impede the viral interaction with the entry receptor ACE2 on the host cell, including S-specific neutralization antibodies (Abs)[13–25] and rhACE2[26–30], inhibit infectivity and prevent COVID-19. Although many high-affinity monoclonal antibodies were identified from convalescent patients and engineered as therapeutics to treat mild diseases of COVID-19[13–25], many did not show favorable efficacy for hospitalized patients[31,32] and some of those with emergency use authorization (EUA) lost efficacy against new variants, such as δ[33]. Monoclonal antibodies targeting specific epitopes of SARS-CoV-2 antigens appear to have limited capacity to broadly neutralize current and future mutant strains[4–6]. Nonetheless, since the plasma or sera of convalescent COVID-19 patients have reportedly been used to treat active infection of SARS-CoV-2 or severe diseases[34,35], we aimed to identify previously unknown anti-viral components from the human plasma that may inform on potential new therapeutics.

Extracellular vesicles (EVs) are one of the essential components of liquid biopsy such as blood, including large microvesicles (200–1000 μm), small exosomes (50–200 μm), and newly identified exomeres (<50 μm)[36,37]. Exosomes are amongst the best characterized small EVs that likely participate in a variety of physiological and pathobiological functions[38–42] as well as serve as novel biomarkers and next-generation biologic therapeutics[43,44]. They present many proteins on the surface reminiscent of their cellular counterpart, such as immune regulators of myeloid and lymphoid cells to affect antiviral immune response[42,45,46]. Exosomes derived from both plants and human specimens have been used in clinical trials to treat inflammatory diseases and cancers[47–49]. In line with widely adopted nomenclature in the EV field[37] and the possibility that heterogenous vesicle populations may be isolated, we collectively refer to the enriched exosomes therein as 'EVs'.

Here we detected a significant increase in circulating ACE2+ EVs in the plasma of COVID-19 patients, in particular during the acute phase. Importantly, ACE2+ EVs (evACE2) isolated from engineered cell lines inhibit SARS-CoV-2 infection by blocking the viral spike protein binding with its cellular receptor ACE2 in host cells. Our observations demonstrate that evACE2 is a decoy antiviral mechanism to prevent SARS-CoV-2 infection, thus providing a rationale for the use of evACE2 to combat COVID-19.

## Results

**Circulating evACE2 increased in the peripheral blood of COVID-19 patients.** We previously established an automated and high throughput method, microflow vesiclometry (MFV), to detect and profile the surface proteins of blood EVs at single-particle resolution[43]. Direct MFV analysis of circulating EVs in human plasma samples (Table 1, $N = 89$) revealed elevated ACE2+ EVs in the plasma of COVID-19 patients in comparison to seronegative controls, with a more dramatic increase in the acute phase and a modest elevation in the convalescent-phase (Fig. 1a, b, Supplementary Fig. 1a–d), with the latter in association with COVID-19 severe disease showing relatively higher levels in inpatient samples (Supplementary Fig. 1b). ACE2+ EVs were enriched in CD63+ EV subsets from COVID-19 patients

**Table 1 Summary of sero-negative, acute, and convalescent-phase of COVID-19 patients from which the plasma ACE2+ EVs and RBD-IgG levels were measured.**

|  | Sero-negative (N = 5) | | CSB convalescent (N = 61) | | CBB acute phase cohort (N = 23) | |
|---|---|---|---|---|---|---|
|  | Mean/count | SD (%) | Mean/count | SD (%) | Mean/count | SD (%) |
| Age, years | 39.7 | 15.9 | 42.9 | 15.0 | 61.5 | 15.4 |
| Sex, male | 0 | 0.0% | 23 | 36.9% | 13 | 56.5% |
| *Race* |  |  |  |  |  |  |
| Black | 0 | 0.0% | 6 | 9.4% | 9 | 39.1% |
| White | 3 | 60.0% | 37 | 57.8% | 9 | 39.1% |
| Asian | 0 | 0.0% | 4 | 6.3% | 1 | 4.4% |
| Other | 2 | 40.0% | 17 | 26.6% | 4 | 17.4% |
| SOFA score[a] | NA (N = 0) | NA | 6.0 (N = 4) | 3.4 | 8.6 (N = 14) | 4.1 |
| *Intermediate to advanced interventions* |  |  |  |  |  |  |
| Vasopressor use | 0 | 0.0% | 4 | 6.3% | 10 | 43.5% |
| High flow nasal cannula | 0 | 0.0% | 2 | 3.1% | 12 | 52.2% |
| Non-invasive ventilation | 0 | 0.0% | 1 | 1.6% | 3 | 13.0% |
| Mechanical ventilation | 0 | 0.0% | 3 | 4.7% | 13 | 56.5% |
| *Length of stay (LOS, days)* |  |  |  |  |  |  |
| ICU patients | NA (N = 0) | NA | 22.4 (N = 5) | 16.1 | 31.3 (N = 15) | 16.7 |
| Inpatients | NA (N = 0) | NA | 5.1 (N = 7) | 3.0 | 9.5 (N = 8) | 5.7 |
| Onset to sampling, days | NA | NA | 87.5 | 47.8 | 12.9 | 9.4 |

[a]The SOFA score was calculated on ICU patients only. All patients are alive at the time of preparing this manuscript.

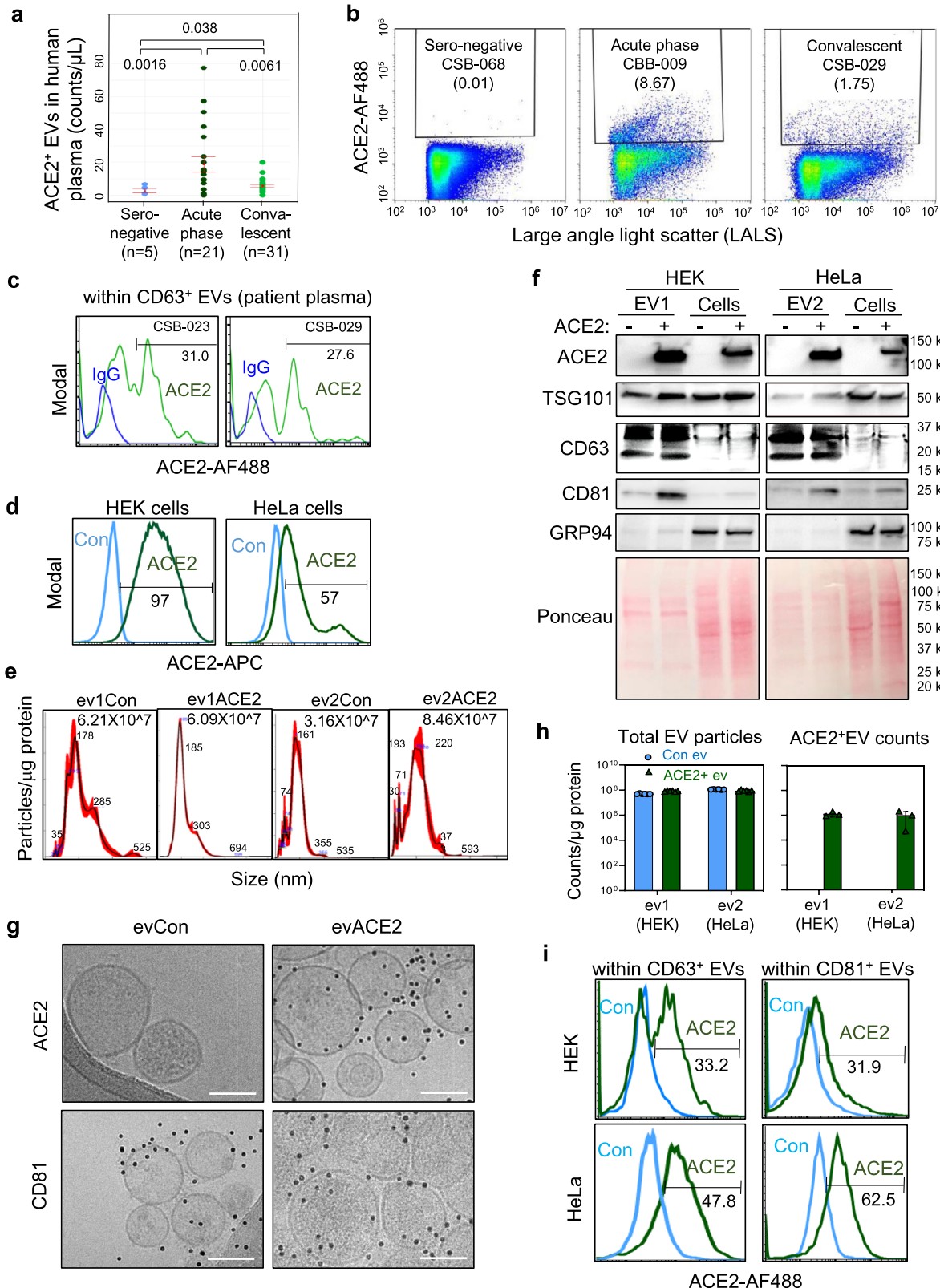

(Fig. 1c). Consistently, SARS-CoV-2 infection triggered secretion of ACE2$^+$TSG101$^+$ EVs by human pneumocyte A549 cells overexpressing ACE2 (Supplementary Fig. 1e), implying that upregulated production of ACE2$^+$ EVs is part of the innate response to SARS-CoV-2 infection in COVID-19 patients.

In order to elucidate the specific functions of evACE2 in anti-SARS-CoV-2 infection, we established a working protocol

for characterizing ACE2 expression in EVs and determining the binding activity and neutralizing functions of evACE2 to SARS-CoV-2.

We generated two sets of human cell lines HEK-293 (HEK) and HeLa, originally negative for ACE2 (ACE2$^-$ control), with stable expression of ACE2 (Fig. 1d). We then utilized a standard ultracentrifugation protocol ($100,000 \times g \times 70$ min) to isolate

**Fig. 1 Circulating evACE2 increased in the peripheral blood of COVID-19 patients. a** ACE2+ EVs detected in human plasma samples of sero-negative controls (light blue), acute phase (dark green), and convalescent COVID-19 patients (green). One-tail $t$ test (*$p = 0.038$, **$p = 0.0061$ and **$p = 0.0016$). Data are presented as mean values ± SEM. **b** Representative microflow vesiculometry (MFV) plots with gated ACE2+ EVs from sero-negative, acute phase and convalescent COVID-19 patients. **c** MFV detection of circulating ACE2$^+$ EVs with CD63$^+$ EVs in human plasma of convalescent COVID-19 patient samples (CSB-029 and CSB-023) (green line). Blue line is isotype IgG-negative control. **d** Flow profiles of ACE2 expression in HEK and HeLa parental control cells (Con, light blue line, ACE2$^-$) and with ACE2 overexpression (ACE2, green line). **e** NanoSight NTA analysis of the sizes of HEK-derived ACE2$^-$ (ev1Con) and ACE2$^+$ (ev1ACE2) and HeLa-derived ACE2$^-$ (ev2Con) and ACE2$^+$ (ev2ACE2). **f** Immunoblots of HEK and HeLa (ACE2$^-$ and ACE2$^+$) EVs and cell lysates for ACE2, TSG101, CD63, CD81, GRP94 and loading control of the membrane proteins upon Ponceau staining. RIPA buffer and Bradford protein assay were used for cells/EVs lysis and protein measurement, respectively ($N = 1$ experiment). **g** Cryo-EM images of HEK-derived EVs, ACE2$^-$ (evCon, left) and ACE2$^+$ (evACE2, right), stained with ACE2 (top) and CD81 (bottom). Scale bars = 100 nm. **h** Quantified counts of Apogee MFV-based total extracellular vesicles (EVs) and ACE2$^+$ EVs ($N = 2$ experiments with $n = 6$ technical replicates for total EV particles and $n = 3$ technical replicates for ACE2$^+$ counts). Control EVs are in light blue and ACE2$^+$ EVs in green. Data are presented as mean values +/− SD. **i** Overlay flow profiles of ACE2 positivity within CD63$^+$ (left column) and CD81+ (right column) EVs isolated from HEK-ACE2 (top row) and HeLa-ACE2 (bottom row) cells, respectively ($n = 3$ technical replicates). Light blue line for Control EVs and green line for ACE2$^+$ EVs.

exosome-enriched EVs from the culture supernatants of these cells after removal of cell debris and apoptotic bodies (Supplementary Fig. 2a). Using nanoparticle tracking analysis (NTA), EVs purified from ACE2$^+$ and control cells exhibited a mean size of ~180–200 nm with equivalent vesicle counts of $6–8 \times 10^7$ per µg of EV proteins (Fig. 1e). Immunoblotting demonstrated that the EVs from ACE2-expressing cells, but not the control EVs, were positive for ACE2 while both EVs displayed exosome-enriched markers CD63, CD81, TSG101, and Syntenin-1 (a newly identified high-abundance exosome marker[50]), and lacked expression of the endoplasmic reticulum protein marker GRP94 (Fig. 1f, Supplementary Fig. 2b–e). We also confirmed that evACE2 purification via ultracentrifugation and/or Optiprep density gradient fractionation did not detect any His-tagged soluble ACE2 (extracellular region) or cleaved ACE2 (which would have relatively smaller molecular weight compared to evACE2 that contains a transmembrane domain), when His-tagged recombinant human ACE2 extracellular domain (rhACE2) was spiked to the culture supernatant prior ultracentrifugation (Supplementary Fig. 2b, c), or to precipitated EVs prior to Optiprep fractionation (Supplementary Fig. 2d, e). Lack of detectable His-tagged rhACE2 in purified EVs indicates that EV purification does not accumulate appreciable soluble ACE2. The full-length ACE2 was almost exclusively detected in small EV fractions co-expressing CD81, with minimal or no detectable ACE2 in the non-vesicular fractions that express the putative exomere marker HSP90 (Supplementary Fig. 2d, e). These results indicate that evACE2 is enriched in small EVs with minimal detection in presumed exomeres.

We further developed high-resolution cryogenic electron microscopy (cryo-EM) along with the high-throughput MFV to analyze ACE2 expression on EVs at single-particle resolution. Both methods detected ACE2 in the EVs derived from ACE2$^+$ HEK and/or HeLa cells, but not from their parental ACE2$^-$ controls, whereas almost equivalent numbers of total EVs ($0.5 \sim 1 \times 10^8$ counts per µg EV proteins) were produced from these cells (Fig. 1g, h, Supplementary Fig. 3a–c), consistent with the NTA analyses. Immuno-cryo-EM revealed distinct expression of ACE2 (~52%) in ACE2$^+$ HEK cell-derived spheric EVs positive for CD81 (Supplementary Fig. 3b). Double staining MFV analyses detected ACE2 in CD81$^+$ EVs (31.9–62.5%) or CD63$^+$ EVs (33.2–47.8%) for HEK-ev1 and HeLa-ev2 (Fig. 1h, i). We subsequently quantified the average ACE2 concentrations or molecular ratios in evACE2 utilizing ELISA and immunoblotting analyses with rhACE2 as a standard. Both methods detected a similar range of ACE2 content in the isolated EVs, including 0.1–0.2 ng ACE2 per µg EV protein of HEK-ev1 and HeLa-ev2 (EV protein measured in PBS via Nanodrop) (Supplementary Fig. 3d–f). Based on the molecular weight of ACE2 and the

number of EV particles detected in isolated HEK-ev1 and HeLa-ev2 respectively, each EV might present 20–40 ACE2 molecules. Collectively, our results demonstrate that the SARS-CoV-2 entry receptor ACE2 protein is expressed on EVs, most likely as a full-length transmembrane protein.

**evACE2 blocks SARS-CoV-2 RBD binding and variant infections.** To analyze the effects of evACE2 on viral attachment and infection, we implemented a flow cytometry-based assay assessing the SARS-CoV-2 S protein (RBD)-binding to human host cells (Fig. 2a, Supplementary Fig. 4a). As expected, ACE2$^+$ HEK cells displayed a specific and robust binding (>90%) with a red fluorophore AF-647-conjugated RBD protein (Supplementary Fig. 4b, d). In a dose-dependent manner, the cell-bound RBD probe signals, both percentages of AF-647$^+$ cells and the mean fluorescence intensity (MFI) were significantly inhibited by pre-incubation with 5 µg of ACE2$^+$ EVs (0.5–1.0 ng evACE2) (Fig. 2b, c, Supplementary Fig. 4c). In contrast, an equal amount of ACE2$^-$ EVs (5 µg) had negligible effects (Supplementary Fig. 4c), indicating that the ACE2$^+$ EVs inhibit SARS-CoV-2 RBD recognition with their cellular receptor ACE2 through decoy ACE2 on EVs. As a positive control, rhACE2[26,29] (140 ng) also inhibited the RBD binding to human ACE2$^+$ cells (Fig. 2b, Supplementary Fig. 4c).). Based upon evACE2 and rhACE2 serial dilution-mediated RBD neutralization assays, the IC$_{50}$ values of evACE2 are 77.06 and 87.16 pM for ev1ACE2 and ev2ACE2 from ACE2 overexpressing HEK cells and Hela cells, respectively, whereas the IC$_{50}$ for soluble rhACE2 to inhibit RBD binding to host cells is 10.37 nM (Fig. 2c). Therefore, evACE2 possesses 120–135 times more efficient blocking of SARS-CoV-2 viral RBD binding to human host cells than soluble rhACE2.

Next, we evaluated the neutralization effects of evACE2 and rhACE2 on the infectivity by SARS-CoV-2 and its variants. When the SIV3-derived SARS-CoV-2 S$^+$ pseudovirus with either a dual Luc2-IRES-Cherry reporter or a luciferase protein reporter was utilized, ACE2$^+$ EVs (ev1ACE2 and ev2ACE2), instead of ACE2$^-$ control EVs, blocked SARS-CoV-2 S$^+$ pseudovirus infection in a dose-dependent manner as shown by flow cytometry of Cherry expressing cells or by luminescence signal of cellular luciferase activity (Fig. 2d, Supplementary Fig. 5a–h). In comparison to an IC$_{50}$ of 459.50 pM for rhACE2, the IC$_{50}$ values for ev1ACE2 (HEK) and ev2ACE2 (HeLa) were 8.01 and 13.63 pM, respectively, representing an estimated 58- and 34-fold higher neutralization efficacy in blocking pseudovirus infection compared to rhACE2 (Fig. 2d). Preincubation of SARS-CoV-2 S$^+$ pseudovirus with ACE2$^+$ EVs did not yield any infection of ACE2-negative cells given our experimental setup (Supplementary Fig. 5f), limiting the possibility that ACE2$^+$ EVs preincubation would promote SARS-CoV-2 infection in ACE2$^-$ cells.

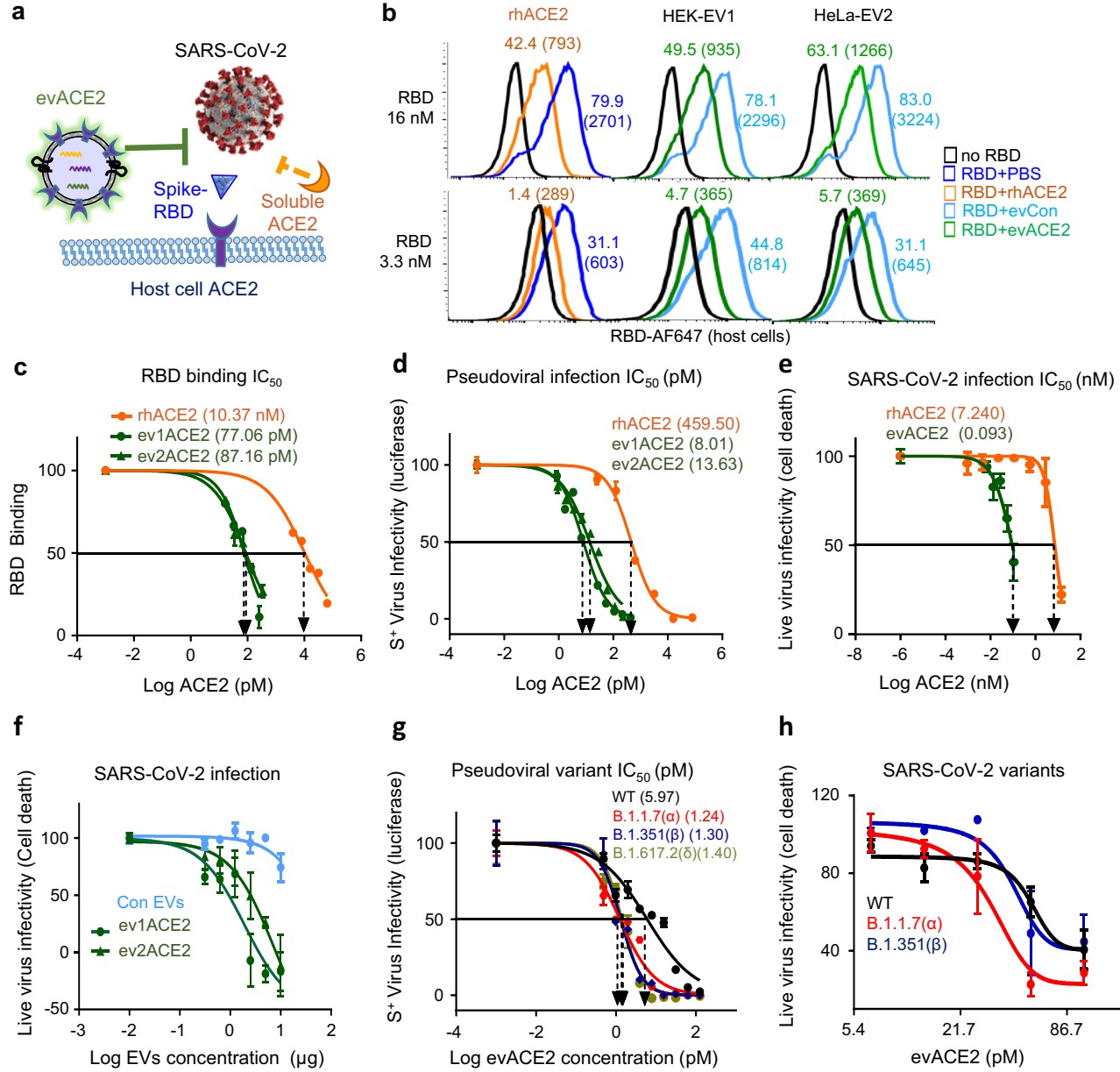

**Fig. 2 Neutralization effects of evACE2 on RBD-binding and SARS-CoV-2 variant infections. a** Schematic depiction of the cell-based neutralization assay. **b** Representative flow profiles showing the percentage (fluorescence mean intensity) of RBD-AF647 binding (at 16 and 3.3 nmol/L) to ACE2+ HEK-293 cells, inhibited by rhACE2 and ACE2+ EVs (evACE2) isolated from HEK-293 and HeLa cells (HEK-EV1 and HeLa-EV2, respectively) whereas ACE2− EVs (evCon) had no neutralization effects (no RBD in black, PBS in dark blue, rhACE2 in orange, evCon in light blue, and evACE2 in green). **c** IC$_{50}$ of rhACE2 (orange line) and ACE2 in the EVs from ACE2+ HEK (ev1ACE2) and HeLa (ev2ACE2) cells (green lines) on 16 nM RBD-host cell binding (%). GraphPad Prism 9.0.2 was used to calculate the IC$_{50}$. $N = 2$ experiments with two technical replicates for each. Data are presented as mean values ± SD. **d** IC$_{50}$ of evACE2, ev1 from HEK and ev2 from HeLa cells (green lines), and rhACE2 (orange line) neutralizing infections by wild-type (WT) S+ pseudotyped SARS-CoV-2. GraphPad Prism 9.0.2 was used to calculate the IC$_{50}$. $N = 2$ experiments with two technical replicates for each. Data are presented as mean values ± SD. **e** IC$_{50}$ (nM) of ACE2 in ev1ACE2 (HEK) (green line) and rhACE2 (orange line) upon wild-type SARS-CoV-2 infection. GraphPad Prism 9.0.2 was used to calculate the IC$_{50}$ with three biological replicates. Data are presented as mean values ± SD. **f** Distinct effects of ACE2+ EVs (green lines) and ACE2− control EVs (light blue line) on inhibiting Vero-6 cell death caused by SARS-CoV-2. $N = 2$ experiments with three biological replicates each. Data are presented as mean values ± SD. **g** The IC$_{50}$ of ev1ACE2 (HEK) neutralizing infections by pseudotyped SARS-CoV-2 expressing WT (black), B.1.1.7 (α) variant (red), B1.351 (β) variant (dark blue) and B.1.617.2 (δ) (light green) S protein. GraphPad Prism 9.0.2 was used to calculate the IC$_{50}$. $N = 2$ experiments with two technical replicates each. Data are presented as mean values ± SD. **h** Effects of ev1ACE2 (HEK) on protecting Vero-6 cell viability against infections of SARS-CoV-2 WT (black), B.1.1.7 (α) variant (red) and B1.351 (β) variant (dark blue) ($n = 3$ biological replicates). Data are presented as mean values ± SD.

We further demonstrated that upon WT SARS-CoV-2 infection (400 plaque-forming units or pfu), the IC$_{50}$ doses for evACE2 (ev1 and ev2) was 41.92–93.65 pM (1.93–4.32 µg EV) in inhibiting the loss of viable Vero-6 cells with decreased viral loads, whereas ACE2− EVs failed to protect the cells (Fig. 2e, f). In comparison to the IC$_{50}$ of rhACE2 at 7.24 nM, evACE2 achieves at least an estimated 80-fold neutralization efficacy to block SARS-CoV-2 viral infections. Consistently,

evACE2-mediated inhibition of SARS-CoV-2 viral loads in infected cells was validated by PCR tests (Supplementary Fig. 5i).

To examine the potential of evACE2 in neutralizing SARS-CoV-2 variants, we utilized both pseudotyped and authentic SARS-CoV-2 infection assays. Importantly, evACE2 achieved up to 4 to 5-fold greater efficacy in blocking the infection of pseudotyped SARS-CoV-2 variants that express S protein mutants from B1.1.7 (α variant), B1.351 (β variant), and B.1.617.2 (δ variant) when compared to WT (Fig. 2g). Similar results were obtained in evACE2-mediated neutralization against authentic SARS-CoV-2 variants, wherein evACE2 inhibited the infection by variants α and β to similar or greater efficacy than WT (Fig. 2h). Collectively, our results support the use of evACE2 as an innovative methodology to prevent or limit infection by a broad spectrum of SARS-CoV-2 viruses, including both WT and variant strains.

**ACE2$^+$ EVs from human plasma neutralize SARS-CoV-2 infection**. Our discoveries that circulating ACE2$^+$ EVs are upregulated in the blood from COVID-19 patients and that the engineered ACE2$^+$ EVs block SARS-CoV-2 infection implies that evACE2 presents with a potential innate antiviral mechanism. We then investigated whether ACE2$^+$ EV abundancy is associated with the viral neutralization effect of human plasma. The elevated RBD-IgG levels in COVID-19 patient plasma were significantly associated with increased neutralization of RBD binding to human cells (Supplementary Fig. 6a, b). Analysis of variance suggested that combining ACE2$^+$ EVs levels with RBD-IgG levels, significantly improved the model fitting to explain the RBD neutralization activity of the tested plasma, with evACE2 accounting for 6.7% of the effects ($p = 0.027$). The multivariable linear regression[51,52] of plasma neutralization activity on both ACE2$^+$ EVs and RBD-IgG levels show an improved $R^2$ of 0.679 (model's goodness-of-fit) than the RBD-IgG levels alone (Supplementary Fig. 6c), supporting a potential antiviral contribution of circulating ACE2$^+$ EVs.

In order to determine the contribution of plasma EVs to neutralization functions, we isolated EVs from seronegative control and COVID-19 patient plasma samples, among which some samples had undetectable or low levels of RBD-IgG (Supplementary Fig. 6d). Following the plasma dilution and extended ultracentrifugation, we detected EVs in the pellets using cryo-EM (Fig. 3a, b). The COVID-19 plasma pellet (acute phase CBB-013 and convalescent CSB-012) showed a transmembrane ACE2 band coupled with positive detection of an exosome marker TSG101 (Fig. 3c, Supplementary Fig. 6e). Mass spectrometry analysis of the RBD-bead pull-down materials from the patient plasma EV pellet confirmed the presence of ACE2 and EV proteins (Supplementary Data 1).

Notably, the EV pellets isolated from the plasma of 5 of 6 acute phase COVID-19 patients, especially CBB-007 and CBB-012 (no detectable RBD-IgG) completely blocked SARS-CoV-2 infection-caused cell death (Fig. 3d, Supplementary Fig. 6d). In contrast, seronegative control and CBB-005 without detectable ACE2 EVs did not show any virus neutralization effects (Fig. 3c, d), suggesting that the plasma ACE2$^+$ EVs levels, potentially regulated by SARS-CoV2 infection, represent a potent antiviral function in suppressing infection by SARS-CoV-2.

We then used RBD-conjugated magnetic beads to deplete the majority of ACE2$^+$ EVs in the plasma pellets (Fig. 3e), some of which had minimal or absent RBD-IgGs prior to and after depletion such as CSB-024 (Supplementary Fig. 6f). Importantly, depletion of ACE2$^+$ EVs isolated from five plasma samples (convalescent CSB-012 and CSB-24, and acute phase CBB-008, 009 and 013) significantly impaired the ability of plasma EVs to neutralize RBD-binding to ACE2$^+$ HEK cells (Fig. 3f), indicating that the ACE2$^+$ EVs in the plasma from COVID-19 patients were at least partially responsible for the anti-SARS-CoV-2 activity.

**Intranasal evACE2 protects hACE2 mice from SARS-CoV-2-caused mortality**. To further validate our discovery of evACE2 as a decoy therapy to treat COVID-19, we evaluated its preclinical therapeutic efficacy using a well-established hACE2 transgenic COVID-19 mouse model[30,53,54]. In our study, the hACE2 transgenic mice showed acute weight loss 2–5 days following intranasal SARS-CoV-2 infection (Fig. 4a). While recovery was often observed from day 6 to 7, and with a full recovery in about 2 weeks after a low dose of SARS-CoV-2 infection, hACE2 mice had high mortality with a high dose of viral infection due to severe lung injury. Within a week after infection with 10,000 pfu of SARS-CoV-2, nearly all mice succumbed (with 20% body weight loss, see the "Methods" section) when treated with control EVs (Fig. 4a). Treatment with nasally delivered ACE2$^+$ EVs (130 μg/mouse) significantly protected 80% of hACE2 mice from SARS-CoV-2 infection-induced mortality (Fig. 4a). The protective activity of evACE2 could be due to their inhibition of SARS-CoV-2 infection of lung epithelial cells, given a reduction in the SARS-CoV-2 viral load detected in lung tissues from hACE2 mice treated with evACE2 compared to control EVs (Fig. 4b).

SARS-CoV-2 infection of hACE2 mice resulted in lung injury that mimicked human COVID-19 pathogenesis, with histopathology consisting of interstitial pneumonia with infiltration of considerable numbers of macrophages and lymphocytes into the alveolar interstitium, and accumulation of macrophages in alveolar cavities[55–57]. This COVID-19 lung pathogenesis was captured in H & E staining analysis of the lung tissue sections from the EV control group of hACE2 mice infected with SARS-CoV-2 (Fig. 4c). Consistent with the reduced viral load in evACE2 treated mice, double-blind pathological scoring revealed that evACE2 treatment largely diminishes lung inflammation in the mice infected by SARS-CoV-2 (Fig. 4d). Consequently, evACE2 treatment effectively protected hACE2 mice from SARS-CoV-2 infection-mediated lung injury, with significantly reduced alveolar hemorrhage and necrosis scores compared to those in control EV-treated mice (Fig. 4e). To further validate whether the nasally delivered ACE2$^+$ EVs are able to neutralize SARS-CoV-2 in mouse lungs, we generated the fluorophore PKH67-labeled ACE2$^+$ EVs and determined their biodistribution. Indeed, when PKH67-labeled ACE2$^+$ EVs were nasally delivered at the therapeutic dosage, their biodistribution was mainly limited to the lungs for local therapy (Supplementary Fig. 6g, h). These results clearly demonstrate that evACE2 achieves a favorable preclinical efficacy to treat COVID-19 pathogenesis.

**Discussion**

Our studies have defined evACE2 as an innovative decoy therapeutic that efficiently blocks the infectious diseases caused by SARS-CoV-2 and its variants of concern, and presumably all future emerging coronaviruses that utilize ACE2 as their initial tethering receptor. Mechanistically, evACE2 inhibits SARS-CoV-2 infection by competing with host cell surface ACE2, which has been also speculated in a recent study showing that rhACE2 inhibits SARS-CoV-2 infection[26,29] and ACE2-containing EVs bind to SARS-CoV-2 spike protein[58,59]. Consistent with the fact that one EV can only carry a limited number of total protein molecules[60], our quantification analysis by cryo-EM, ELISA and immunoblotting estimated up to 20–40 ACE2 molecules per EV. Importantly, evACE2 possesses an 80-fold better efficiency to block SARS-CoV-2 infection than soluble rhACE2. Of note, it has

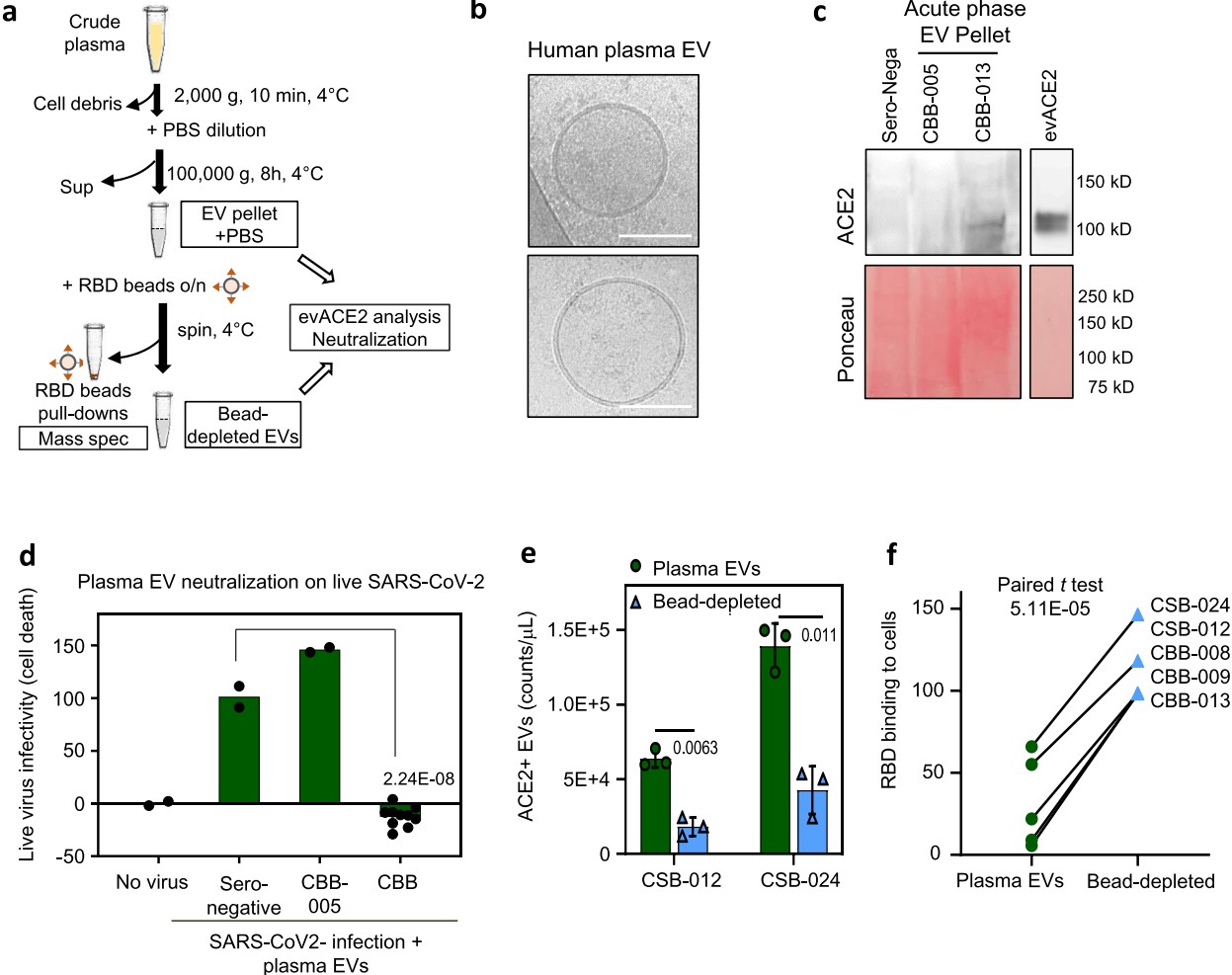

**Fig. 3 evACE2 in patient plasma neutralizes SARS-CoV-2. a** Schematic depiction of plasma EV ultracentrifugation and RBD-bead based depletion. **b** Cryo-EM images of human EV pellets isolated from acute phase COVID-19 plasma (bar = 100 nm). **c** Immunoblots of plasma EV pellets (sero-negative and COVID-19 acute phase patients CBB-005 and -013) for ACE2 and loading control of protein staining with Ponceau. Laemmli buffer was used for lysis (N = 1 experiment). **d** ACE2$^+$ EV pellets from acute phase patients 007, 008, 009, 012, and 013 (CBB) (n = 2 biological replicates each) blocked SARS-CoV-2 infection-induced death of Vero-6 cells whereas the sero-negative control (n = 2 biological replicates) and CBB-005 (no detectable ACE2) (n = 2 biological replicates) did not show neutralization effects. One-tail t test, ****p = 2.24E−08 shown as compared to sero-negative. **e, f** Levels of ACE2$^+$ EV counts (n = 3 biological replicates) in plasma EVs (green) and bead-depleted EVs (light blue). One-tail paired t test, *p = 0.011 and **p = 0.0063 (data are presented as mean values ± SD) (**e**) and altered neutralization effects on RBD–host cell binding (**f**) of the COVID-19 plasma EV pellets prior to and after RBD-bead depletion (convalescent phase CSB-012 and -024; acute phase CBB-008, 009, and 013). One-tail paired t test ****p = 5.11E−05.

been recently reported that the exosomal delivery of STINGa potentiates its uptake into dendritic cells compared with STINGa alone, which led to increased accumulation of activated CD8$^+$ T-cells and an antitumor immune response[61].

Almost all dominant SARS-CoV-2 variants of concern harbor mutations in the RBD of S protein, such as N501Y (α and β) and E484K (β) that facilitate and strengthen the interaction between the virus and ACE2 receptor[1,4,5,10–12]. The delta variant mutations not only result in enhanced receptor binding but also increase the rate of S protein cleavage, resulting in enhanced transmissibility[62]. While such mutations potentially render the variants resistant to vaccine-induced immunity and existing monoclonal antibody therapy, evACE2 can bind and neutralize these variants with an equal or even higher efficacy than for the WT strain, supporting their potential use as a broad-spectrum antiviral mechanism.

While future studies are needed, we speculate the following two potential mechanisms underlying how evACE2 possibly achieves a better efficacy than soluble ACE2 or ACE2-conjugates[63,64] to

block SARS-CoV-2 infection: first, as small EVs are in an average size of 100–200 nm, proteins presented on small EVs might amplify the space interval in suppressing SARS-CoV-2 access to its host cell surface. Second, it is also possible that EV expression may increase the affinity of ACE2 binding with the SARS-CoV-2 S protein through synergy among ACE2 proteins on the same EV, and/or through the transmembrane domain which is involved in presenting an optimal ACE2 conformation for binding with S proteins.

Beyond a significant amplification of evACE2 in the anti-SARS-CoV-2 efficacy in comparison to the purified rhACE2[65], we speculate that the therapeutic efficacy of evACE2 could be further potentiated through co-delivering additional anti-SARS-CoV-2 medicines[66,67]. This integration between surface ACE2 and antiviral medicine may allow us to develop superior therapies as well as reduce the potential side effects from both therapeutics. EVs have been utilized as drug delivery systems with therapeutic potential against various disorders including infectious diseases and cancers[44,68]. Of note, EVs derived from both

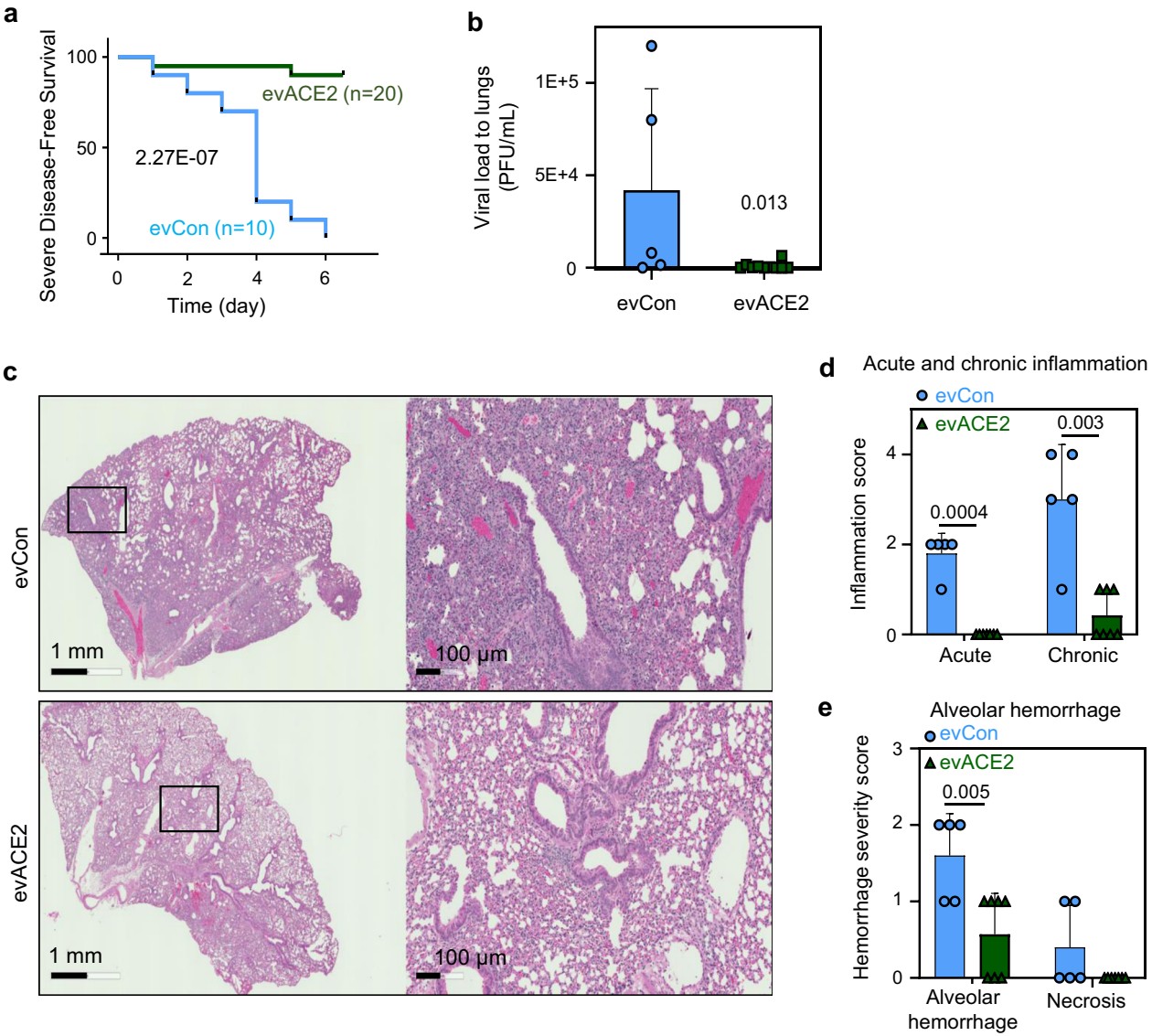

**Fig. 4 evACE2 inhibits SARS-COV-2 infection and inflammation in hACE2 transgenic mice. a** Probability of severe disease-free survival in B6.Cg-Tg(K18-ACE2)2Prlmn/J (K18-hACE2) mice receiving SARS-CoV-2 infection (10,000 pfu) and intranasal EVs (130 μg as measured on Nanodrop) per mouse (evCon in light blue and evACE2 in green). Log-rank (Mantel–Cox) and Gehan–Breslow–Wilcoxon tests ****$p = 2.27E-07$. **b** Viral loads in mouse lungs on day 5/6 after receiving SARS-CoV-2 infection and administration of evCon ($N = 5$ mice) (light blue) or evACE2 ($N = 10$ mice) (green). $T$-test–nonparametric-one tailed, *$p = 0.013$. Data are presented as mean values ± SD. **c** Representative H&E images of mouse lung sections at day 5 or 6 post virus inoculation and EV treatment (evCon and evACE2) intranasally. **d**, **e** Acute and chronic inflammation scores (**d**), and alveolar hemorrhage and necrosis scores (**e**) in mouse lungs on day 5/6 after receiving evCon ($N = 5$ mice) (light blue) or evACE2 ($N = 7$ mice) (green). $T$-test–nonparametric-one tailed, **$p = 0.005$, **$p = 0.003$ and ***$p = 0.0004$. Data are presented as mean values ± SD.

plants and human specimens, such as dendritic cells and tumor cells, have been evaluated in multiple clinical trials and proven safe in humans[47–49]. For example, cancer cell-derived EVs containing chemotherapeutic drugs, in addition to neo-antigens have been used to treat patients with malignant pleural effusion (NCT01854866 and NCT02657460). The EVs derived from plants, including grape (NCT01668849) and ginger or aloe (NCT03493984), have been registered in clinical trials in treating radiation- and chemotherapy-induced oral mucositis.

Circulating EVs in plasma represents an important component of blood in terms of their defensive, homeostatic, and signal transduction properties[42,43,45,46]. Importantly, our discovery reveals that after SARS-CoV-2 infection, a substantial amount of ACE2$^+$ EVs present in human plasma can function as an innate

antiviral mechanism acting as a decoy to protect host cells from coronavirus infection. The levels of this innate antiviral evACE2 appear to be elevated by and in response to acute SARS-CoV-2 infection as shown in the plasma from some COVID-19 patients and in culture media of infected cells. The evACE2 upregulation in the blood appears to be sustained even during the recovery phase from COVID-19 patients with severe disease. It has been well established that the clinical disease severity may be positively associated with higher SARS-CoV-2 virial load[69], implying a possibility that either the virial pathogens or their associated pathogenesis, induce the generation of evACE2. Interestingly, EVs from COVID-19 patients have been reported to play a role in the pathogenesis of the disease[70]. Nevertheless, viral infection may modulate the production and content of EVs[71] and

SARS-CoV-2 has been shown to promote lysosomal exocytosis[72]. Future studies are needed to investigate the molecular mechanisms underlying the regulation of evACE2 production following SARS-CoV-2 infection.

## Methods

**Human subject study and biosafety approvals**. All research activities with human blood specimens of pre-COVID-19, seronegative (healthy) donors, and acute and convalescent COVID-19 patients were implemented under NIH guidelines for human subject studies and the protocols approved by the Northwestern University Institutional Review Board (STU00205299, STU00212371, and STU00205299-MOD0001) as well as the Institutional Biosafety Committee. For collecting human blood specimens, patients and donors were recruited at Northwestern Memorial Hospital based on their availability and willingness to consent and participate in the research. Participants were not compensated.

*Animal study statement*. Experiments with SARS-CoV-2 were performed in biosafety level 3 (BSL3) and animal BSL3 (ABSL3) containment in accordance with the institutional guidelines following experimental protocol review and approval by the Institutional Biosafety Committee (IBC) and the Institutional Animal Care and Use Committee (IACUC) at the University of Chicago. EV biodistribution studies with B6 mice in the absence of viral infections were approved by the Institutional Animal Care and Use Committee (IACUC) at Northwestern University.

**Cell culture**. The parent ACE2⁻ human embryonic kidney HEK-293 cells (HEK) (ATCC, CRL-1573) or human cervical cancer HeLa cells (HeLa) (ATCC, CRM-CCL-2) are transduced with a lentiviral pDual-ACE2 expression vector for stable ACE2 expression and production of ACE2⁺ EVs. Dr. Daniel Batlle and Dr. Jan Wysocki generously provided HEK-293 and HEK-293 cells overexpressing ACE2 (HEK-ACE2). Dr. Thomas Gallagher of Stritch Medical School, Loyola University kindly provided HeLa and HeLa-ACE2 cells via the Hope group. ACE2- parent cells serve as negative controls in the production of ACE2⁺ EVs in the culture. Cells were grown in Dulbecco's Modified Eagle's Medium (DMEM) supplemented with 10% (v/v) fetal bovine serum (FBS), 100 U/mL penicillin and 100 mg/mL streptomycin. FBS used to prepare complete media was EV-depleted by ultracentrifugation at $100,000 \times g$ for 16 h at 4 °C. We used western blotting and flow cytometry analysis to authenticate HEK, HEK-ACE2, HeLa and HeLa-ACE2 cell lines. A549 cells (ATCC, CCL-185) overexpressing hACE2 (A549-hACE2), which were a kind gift of Benjamin TenOever, Mt. Sinai Icahn School of Medicine, and Vero-6 cells (ATCC, CRL-1586) were maintained in DMEM with 10% FBS, 1% penicillin/streptomycin and 1% non-essential amino acids. Cells were tested for mycoplasma contamination before culturing in all the laboratories.

**Flow cytometry**. Cells were blocked with mouse serum IgG (Sigma, 15381) for 10 min at room temperature and then incubated with specific antibodies; AF-647 mouse anti-human ACE2 (Clone # 535919) (R&D systems, FAB9332R), AF-488 mouse anti-human ACE2 (Clone # 171607) (R&D systems, FAB9333G) (0.4 μg/10⁶ cells), AF-647 isotype control mouse IgG2b (Clone # 20102) (R&D systems, IC003R) or AF-488 isotype control mouse IgG2bAF488 (Clone # 20102) (R&D systems, IC003G) for 45 min on ice, followed by washing twice with 2% EV-free FBS/PBS. Finally, the cells were diluted in 2% EV-free FBS/PBS and analyzed on a BD-LSR II flow cytometer (BD Biosciences). Data were analyzed by BD FACSDiva softwares v8.0.2 or v8.0.3 or Flow Jo v10.6.2.

**Isolation of cell culture-derived EVs**. EVs were isolated from the cell culture supernatant of each of the four cell lines[43]. Cells were cultured as monolayers for 48–72 h under an atmosphere of 5% CO₂ at 37 °C. When cells reached a confluency of ~80–90%, culture supernatant was collected, and EVs were isolated using differential centrifugation. First, the supernatant was centrifuged at $2000 \times g$ for 10 min then at $10,000 \times g$ for 30 min to remove dead cells and cell debris. Next, the supernatant was ultracentrifuged for 70 min at $100,000 \times g$ using SW41 Ti or SW32 Ti swinging bucket rotor (Thermo Fisher Sorvall wX+ 80 or Beckman Coulter Optima XE) to pellet the EVs. EVs were then washed by resuspension in sterile PBS (Hyclone, UT, USA), and pelleted by ultracentrifugation for 70 min at $100,000 \times g$. The EV pellet was resuspended in 100 μL PBS and stored at −80 °C. The EV proteins in PBS were measured on Nanodrop in most of the experiments unless specified in certain immunoblotting experiments in Fig. 3c.

**Spiked soluble ACE2 analysis in EV purification**. HEK293 cells overexpressing ACE2 and the parental cells negative for ACE2 were cultured for 48 h in DMEM with 10% exosome-depleted FBS and 1% penicillin–streptomycin. 30 mL of conditioned media was collected from the cells and spiked with 2 μg recombinant ACE2 protein (RayBiotech 23030165–100), followed by incubation at room temperature for 1 h. EVs were isolated from the media as described above.

**Density gradient fractionation of EVs**. EVs were isolated by ultracentrifugation as described above in the section "Isolation of cell culture-derived EVs",

resuspended in PBS, mixed with 2 μg recombinant ACE2 protein (RayBiotech 230-30165-100), and subjected to density gradient fractionation as in ref. [73]. Resuspended pellets were loaded into 2.4 mL of 36% Optiprep (Sigma-Aldrich), followed by sequential layering of 2.4 mL of 30%, 24%, 18%, and 12% Optiprep on top. Samples were then ultracentrifuged at $120,000 \times g$ for 15 h at 4 °C in an SW41 Ti rotor (Beckman Coulter). 1 mL fractions were collected, with numbering starting from the top fraction, diluted with 11 mL of PBS, and washed at $120,000 \times g$ for 4 h at 4 °C in an SW41 Ti rotor. Fractions were then resuspended in the appropriate buffer for downstream analysis. Assuming all rhACE2 is recovered in each density fraction, ~166 ng of rhACE2 is present in each fraction.

**Immunoblotting**. In Figs. 1f and 3c, cells and cell-derived EVs (HEK and HeLa) were lysed using RIPA buffer with protease inhibitor cocktail (Thermo Scientific, 1861279) (1:100 dilution) for 30 min on ice, then centrifuged for 15 min at 4 °C and $18,800 \times g$. Protein was measured using Bradford protein assay (BioRad, 5000006), and 10–20 μg of cell-derived proteins and 2–8 μg of EV-derived proteins (equivalent to 20–80 μg EV proteins measured in PBS via Nanodrop) were denatured at 100 °C for 5 min and loaded to SDS–PAGE (4–20% gels), then transferred to PVDF membranes that were incubated O/N with the primary antibodies. Membranes were then washed, incubated with the corresponding horseradish (HRP)-conjugated antibodies, washed, and then developed using Pierce ECL2 solution (Thermo Fisher Scientific, 1896433A), BioRad ChemiDoc imaging System was used to collect data (Fig. 1f). Human plasma and plasma-derived EV samples (resuspended in PBS) were lysed with Laemmli buffer (Bio-Rad, 1610747) for 30 min on ice and processed as mentioned above. We acknowledge that the amount of total protein in EVs might be read differently via Nanodrop or other protein measurement methods.

In Supplementary Fig. 2c, e, EVs were lysed by urea buffer (8 M urea, 2.5% SDS) with PhosSTOP (Roche 04906845001) and Complete mini EDTA free protease inhibitor (Roche 11836170001) on ice for 30 min. EVs-derived proteins (equivalent to 500 μg total EV proteins as measured on Nanodrop in PBS prior to lysis, loaded per lane) and recombinant ACE2 proteins (100–500 ng loaded per lane, as described in the figures) were denatured with LDS sample buffer (Thermo Fisher NP0007) and DTT at 70 °C for 10 min and loaded to a 4–12% precast polyacrylamide gels, then transferred to a PVDF membrane using the Trans-Blot Turbo transfer system (BioRad). Membranes were blocked with 5% non-fat dry milk at room temperature for 1 h and then incubated with primary antibodies (diluted in 2% BSA) at 4 °C overnight. Membranes were then washed three times with TBS-T (TBS with 0.01% Tween-20) and incubated with secondary antibodies (diluted in 2% non-fat dry milk) at room temperature for 1 h. After washing three times with TBS-T, the membranes were developed using West-Q Pico ECL reagent (GenDepot W3652020) or Pierce ECL reagent (Thermo Fisher, 32106). For Supplementary Fig. 2c, blots were probed for ACE2, washed with TBST, then probed for syntenin-1. In Supplementary Fig. 2e, blots were probed for ACE2, washed with TBST, followed by probing for CD81, or His-tag followed by HSP90.

The antibody dilutions are shown in Supplementary Table 1.

**Detection of ACE2 in ACE2⁺ EVs by ELISA and immunoblotting**. EVs were isolated as mentioned above, lysed using RIPA buffer with protease inhibitor cocktail (1:100 dilution) for 45 min on ice, then centrifuged for 15 min at 4 °C and $18,800 \times g$. First: Human ACE-2 ELISA kit (RayBiotech, ELH-ACE2-1) was used to detect ACE2. The antibody pair detects the extracellular domain of Human ACE-2. The kit was used per the manufacturer's instruction and optical density was measured using BioTek Synergy HT. Second: 27.3 and 87.5 μg of ACE2⁺ EVs (as measured in PBS by Nanodrop) were denatured at 100 °C for 5 min and loaded to SDS–PAGE (4–20% gels), then transferred to nitrocellulose membranes that were incubated O/N with the ACE2 primary antibody (R&D systems, AF933). Membranes were then washed, incubated with the HRP-conjugated antibody, re-washed then detected by Pierce ECL2 solution. BioRad ChemiDoc imaging System was used to collect data and Image Lab 6.1 was used for densitometry quantification (Supplementary Fig. 3d–f).

**Nanoparticle-tracking analysis**. Analysis was performed at the Analytical bio-NanoTechnology Core Facility of the Simpson Querrey Institute at Northwestern University. All samples were diluted in PBS to a final volume of 1 ml and ideal measurement concentrations were found by pre-testing the ideal particle per frame value. Settings were according to the manufacturer's software manual (NanoSight NS3000).

**MFV analysis of EVs**. Antibody solutions were centrifuged at $14,000 \times g$ for 1 h at 4 °C to remove aggregates before use. EVs (1–2 μg EV proteins, as measured on Nanodrop, in 20 μL of PBS) were blocked using 1 μg of mouse serum IgG for 10 min at RT then incubated with: AF-488 mouse anti-human ACE2 (Clone # 171607) (R&D systems, FAB9333G, 0.4 μg/2 μg EVs), APC mouse antihuman CD81 (Clone JS-81 (RUO)) (BD Biosciences, 561958, 1 μL/2 μg EVs), AF-647 mouse antihuman CD63 (Clone H5C6 (RUO)) (BD, Biosciences, 561983, 2 μL/2 μg EVs), AF-488 isotype control mouse IgG2b (Clone # 20102) (R&D systems, IC003G, 0.4 μg/2 μg EVs), APC isotype control mouse IgG₁κ (Clone MOPC-21 (RUO)) (BD Biosciences, 555751, 1 μL/2 μg EVs) or AF-647 isotype control mouse

IgG$_1$κ (Clone MOPC-21 (RUO)) (BD, Biosciences, 557714, 2 μL/2 μg EVs) for 45 min at 4 °C. The solution was then diluted to 200 μL with PBS and the samples were run on Apogee A50 microflow cytometer (MFC) (Apogee Flow Systems, Hertfordshire, UK) (http://www.apogeeflow.com/products.php). The reference ApogeeMix beads (Apogee Flow Systems, 1493), were used to assess the performance of Apogee MFC and to compare the size distribution of the EVs. PBS was run as a background control. Data were analyzed using Flow Jo v10.6.2.

**Immuno-cryo-EM imaging.** Antibody solutions and other staining buffers were centrifuged to remove non-specific particles or aggregates in the buffer of interest, at 14,000 × g for 1 h at 4 °C before use. EVs (10 μg in 100 μL PBS as measured on Nanodrop) were blocked using 5 μg of mouse serum IgG for 10 min at RT then incubated with mouse anti-human ACE2 (R&D systems, FAB9333G), mouse antihuman CD81 (BD Biosciences, 551108), isotype control mouse IgG2b (R&D systems, IC003G) or isotype control mouse IgG$_1$κ (BD Biosciences, 551954) for 45 min at 4 °C. To rinse samples, 1 mL PBS was added to the tubes, and EVs were centrifuged 100,000 × g for 30 min at 4 °C. PBS was aspirated, samples were reconstituted in 100 μL PBS and incubated with EM goat anti-mouse IgG (H&L) 10 nm gold conjugated (BBI solutions, EM.GMHL10) (7:100) for 30 min at RT. EVs were then rinsed by adding 1300 μL PBS then centrifuged at 100,000 × g for 15 min at 4 °C. Finally, PBS was aspirated, and EVs were reconstituted in 50 μL PBS.

For cryoEM visualization, samples were prepared from freshly stained EVs at the concentration provided. For cryo-freezing, 3.5 μL of EV solutions were applied to fresh glow-discharged (10 s, 15 mA; Pelco EasiGlow) lacey carbon TEM grids (Electron Microscopy Services) and vitrified using an FEI Vitrobot Mark IV (FEI, Hillsboro, OR). The sample was applied to the grid and kept at 85% humidity and 10 °C. After a 10 s incubation period, the grid was blotted with Whatman 595 filter paper for 4 s using a blot force of 5 and plunged frozen into liquid ethane. Samples were imaged using a JEOL 3200FS electron microscope equipped with an omega energy filter operated at 200 kV with a K3 direct electron detector (Ametek) using the minimal dose system. The total dose for each movie was ~20 e⁻/A² and was fractionated into 14 frames at a nominal magnification between 8000 and 15,000 (pixel size on the detector between 4.1 and 2.2 Å, respectively). After motion correction of the movies[74], EVs were identified manually using ImageJ[75]. Two grids were prepared and imaged with 10–20 fields for each condition.

**Development of the SARS-Cov-2 RBD "bait".** RBD of 223 amino acid (Arg319-Phe541) fragment of the SARS-CoV-2 Spike protein that binds to the ACE2 receptor (Raybiotech, 230-30162-100) was biotinylated using NHS-PEG4-Biotin (Thermo Fisher, 21330). The protein was de-salted using Zeba Quick Spin columns (Thermo Fisher, 89849) and incubated with Streptavidin-AlexaFluor-647 (SA-AF-647) (Thermo Fisher, S21374) to make the RBD-biotin-AF647 bait.

**Cell-based RBD binding neutralization by ACE2+ EVs and human plasma.** The RBD-biotin-AF647 bait (3.3 and 16 nM) was incubated with EVs (ACE2+ and ACE2−), recombinant human ACE2 extracellular region (rhACE2, RayBiotech, 230-30165-100), or human plasma (10 μL or 80 μL) (creating "neutralized RBD"), then incubated with ACE2+ HEK-293 cells (200,000 cells in 100 μL 2%EV-free FBS/PBS) for 45 min on ice. Human recombinant ACE2 protein was used as a positive control (70–140 ng as determined by ELISA). RBD bait that was incubated with PBS, or with ACE2− EVs, non-fluorescent RBD bait (mock control) and ACE2− cells were used as controls. Cells were then spun and washed twice with PBS. DAPI was added to exclude dead cells analyzed on flow cytometer (BD FACSAria SORP or BD-LSR II flow cytometer (BD Biosciences)) and viable singlets were gated for percentage and mean fluorescence intensity (MFI) measurements of the RBD-AF647+ population. Data were analyzed by BD FACSDiva softwares v8.0.2 or v8.0.3 or Flow Jo v10.6.2.

**Neutralization effects of ACE2+ EVs on SARS-CoV-2 spike+ pseudovirus infection.** The SARS-CoV-2 spike (S+) pseudovirus carrying the Luc2-Cherry reporters was made for live virus neutralization assay after the pcDNA3-spike expression vector was transfected along with pCMV-Luc2-IRES-Cherry and pSIV3+ lentiviral vectors into a lentivirus producing cell HEK-293. Spike B.1.1.7 (α) variant (BPS Bioscience, 78112), B.1.351 (β) variant (BPS Bioscience, 78142) and Spike B.1.617.2 (δ) variant (BPS Bioscience, 78215) pseudotyped lentivirus (Luc Reporter) were used. The S+ pseudovirus and/or variants were incubated with ACE2+ EVs, or ACE2− EVs, or a positive control rhACE2, or negative control (PBS), for 1 h at 37 °C prior to the infection with ACE2+ human host cells HeLa in 96-well plates (5000 cells/well). A bald virus without spike expression and ACE2− cells served as negative controls. Cells were incubated and monitored by the IncuCyte live-cell imaging system (Essen BioScience). Flow cytometry of Cherry and luciferase activity analysis (Promega, EL500) were used to assess viral infectivity (BD-LSR II flow cytometer (BD Biosciences) and BioTek Synergy HT).

**WT and variant SARS-CoV-2 live virus infection to Vero-6 or A549 cells (BSL3).** The WT SARS-CoV-2 live virus study was conducted at the NIAID-supported BSL-3 facility at the University of Chicago Howard T. Ricketts Regional Biocontainment Laboratory. SARS-CoV-2 (nCoV/Washington/1/2020) was

provided by the National Biocontainment Laboratory, Galveston, TX. The strain 2019-nCoV/IDF0372/2020 (WT) was supplied by the National Reference Centre for Respiratory Viruses hosted by Institute Pasteur (Paris, France). B.1.1.7: hCoV-19/France/IDF-IPP11324/2021 and B.1.351: hCoV-19/France/PDL-IPP01065/2021 both kindly supplied through the European Virus Archive goes Global (EVAg) platform.

One day prior to viral infections, 10,000 Vero-6 cells were seeded per well in triplicates onto 96-well plates. 16 h after seeding, the attached cells were infected with mock controls (no virus) and WT SARS-CoV-2 (400 pfu) viruses which were pre-mixed with a serial of doses of EVs (starting from 20 μg with 6 times of 1:2 dilutions) or untreated control. 96 h later, the host cell viability (opposite to viral infectivity-caused cell death) was measured by crystal violet staining which stained attached viable cells on the plate following fixation. Cells killed off by the virus were floating and excluded. For the untreated control, the cells were infected but left without any treatment with a value of maximal cell death caused by the virus. The second control was the mock-infected control where cells grew in the absence of virus or experimental sample representing the maximum normal cell growth over the time period. The absorbance value of the untreated control was subtracted from all other absorbance values, thereby setting untreated wells to "0", then all absorbance values were divided by the mock-infected value thereby making that value 100 (infinite 200Pro).

A549 cells overexpressing ACE2 (A549-hACE2) cells were seeded (25,000 cells/well) in 24-well plates. 16 h after seeding, the attached cells were infected with mock controls (no virus) and WT SARS-CoV-2 (MOI 0.1). Media was collected after 72 h of infection and inactivated at 65 °C for 30 min then shipped to Northwestern University. Media was spun 2000 × g for 10 min, then the supernatant was ultracentrifuged 100,000 × g overnight using SW41 Ti swinging bucket rotor (Thermo Sorwall wX+ 80) for maximal enrichment of evACE2. Pellets were reconstituted in equal volumes of PBS, lysed in RIPA buffer with protease inhibitor cocktail (1:100 dilution) for 45 min on ice, then centrifuged for 15 min at 4 °C and 18,800 × g. EVs were denatured at 100 °C for 5 min and loaded to SDS–PAGE, then transferred to nitrocellulose membranes that were incubated O/N with ACE2 (R&D systems, AF933) and TSG101 (Proteintech, 14497-1-AP) primary antibodies. Membranes were then washed, incubated with the HRP-conjugated antibody, re-washed then detected by Pierce ECL2 solution (Supplementary Fig. 1e).

**RBD-IgG quantitative ELISA assay.** The ELISA protocol was established[76,77] and used herein with the modification of using plasma instead of serum. Plasma samples were diluted by half with PBS during RosetteSep human B cell processing (StemCell Technologies #15064), aliquoted, and stored at −80C until analysis. Plasma was run in quadruplicate and reported as the average (BioTek Synergy HT). Results were normalized to the CR3022 antibody with known affinity to RBD of SARS-CoV-2[78]. Sample anti-RBD IgG concentration reported as μg/mL was calculated from the 4PL regression of the CR3022 calibration curve. A sample value >0.39 μg/mL CR3022 was considered seropositive.

**Plasma EV enrichment by ultracentrifugation.** Sero-negative and COVID-19 (CBB at the acute phase and CSB at convalescent phase) patient-derived plasma samples were obtained from Northwestern Memorial Hospital and stored at −80 °C. Frozen samples were thawed on ice, centrifuged 800 × g for 5 min at 4 °C, and then 2000 × g for 10 min at 4 °C to remove debris. Then the 1 ml plasma supernatant was diluted with 11 mL PBS and ultra-centrifuged at 100,000 × g for 8 h at 4 °C (Beckman Coulter Optima L-90K Ultracentrifuge or Thermo Fisher Sorvall wX+80, SW41 Ti swinging bucket rotor) to isolate and enrich EVs in the pellets. After centrifugation, supernatants and plasma pellets were collected separately. Plasma pellets were resuspended in appropriate volumes of PBS and subject to one round washing and ultracentrifugation at 100,000 × g for 8 h at 4 °C. The levels of ACE2+ EVs in plasma samples were evaluated by MFV on Apogee and western blotting using EV marker TSG101 and ACE2. ACE2+ cell culture-derived EVs were used as a positive control.

**Depletion of ACE2+ EVs by RBD-conjugated beads.** CSB and CBB patient plasma EVs were ultra-centrifuged above, and the EV-enriched pellets were resuspended in 250 μL PBS (per 1 mL plasma) for subsequent bead-mediated depletion. RBD-coupled magnetic beads or anti-ACE2-coupled dynabeads were prepared according to the manufacturer's protocols. Every 25 μL magnetic bead (CELLection Biotin Binder Kit, Thermo Fisher Scientific, 11533D) was coupled with 1 μg of biotin-conjugated RBD protein (ACROBiosystems, SPD-C82E9). EV pellet samples were incubated with the beads for 30 min at 4 °C on a rotator. And then the beads were removed by spinning or magnetic forces. The ratio of plasma samples and RBD-beads is shown in Supplementary Table 2. Beads were removed by magnetic forces. The ACE2+ EV depletion efficiency was confirmed by MFV on Apogee and/or western blotting methods.

The altered neutralization effects of CSB and CBB plasma-derived EVs (resuspended pellets) prior to and after bead depletion was measured via flow cytometry (BD FACSAria SORP) as modified RBD binding to human host cells as described above. And rhACE2 protein (RayBiotech, 230-30165-100) was used as a

positive control (70–140 ng). Data were analyzed by BD FACSDiva software v8.0.3 or Flow Jo v10.6.2.

**LC-MS/MS analysis of RBD-bead precipitated EVs and proteins**. Proteins from RBD-beads precipitated fractions from plasma EV pellets (8 h ultracentrifugation of CSB-012, CSB-024, NWL-001, and NWL-004) and controls of rhACE2 protein and purified ev1ACE2 (from HEK-ACE2 cells) were resuspended in cell lysis buffer (12 mM SDC in 50 mM TEABC with 1% protease and phosphatase inhibitor, pH 8.0), reduced with 10 mM dithiothreitol for 1 h at 25 °C, and subsequently alkylated with 10 mM iodoacetamide for 30 min at 25 °C in the dark. The SDC concentration was diluted 1:4 with 50 mM $NH_4HCO_3$ for enzymatic digestion. Proteins were digested with Lys-C (Wako) and sequencing-grade modified trypsin (Promega, V5117) at 25 °C for 14 h. After digestion, each sample was acidified, desalted, lyophilized, and reconstituted in 12 μL of 0.1% FA with 2% CAN. 5 μL of the resulting sample was analyzed by LC–MS/MS using an Orbitrap Fusion Lumos Tribrid Mass Spectrometer (Thermo Scientific) connected to a nanoACQUITY UPLC system (Waters Corp., Milford, MA) (buffer A: 0.1% FA with 3% ACN and buffer B: 0.1% FA in 90% ACN)[79]. Peptides were separated by a gradient mixture with an analytical column (75 μm i.d. × 20 cm) packed using 1.9-μm ReproSil C18 and with a column heater set at 50 °C. Peptides were separated by a gradient mixture: 2–6% buffer B in 1 min, 6–30% buffer B in 84 min, 30–60% buffer B in 9 min, 60–90% buffer B in 1 min, and finally 90% buffer B for 5 min at 200 nL/min. Data were acquired in a data-dependent mode with a full MS scan ($m/z$ 400–1800) at a resolution of 120 K with the AGC value set at $8 \times 10^5$ and maximum ion injection at 100 ms. The isolation window for MS/MS was set at 1.5 $m/z$ and optimal HCD fragmentation was performed at a normalized collision energy of 30% with AGC set as $1 \times 10^5$ and a maximum ion injection time of 200 ms. The MS/MS spectra were acquired at a resolution of 60 K. The dynamic exclusion time was set at 45 s. The raw MS/MS data were processed with MSFragger via FragPipe[80,81] with LFQ-MBR workflow. A peptide search was performed with full tryptic digestion (Trypsin) and allowed a maximum of two missed cleavages. Carbamidomethyl (C) was set as a fixed modification; acetylation (protein N-term) and oxidation (M) were set as variable modifications. The match-between function has been used based on 1% ion FDR. The final reports were then generated and filtered at 1% protein FDR. The results were shown in Supplementary Data 1 and Source Data. The information of spectral count, ion intensity, and Ion Count obtained via FragPipe was used for label-free quantitation. The cellular components annotation was analyzed by DAVID[82] to compute Fisher's Exact $p$-values. Mass spectroscopy raw data sets have been deposited in the Japan ProteOmeS-Tandard Repository[83] with accession numbers PXD029662 for ProteomeXchange[84] and JPST001379 for jPOST.

**Patient association analyses**. Circulating $ACE2^+$ EV counts, RBD-IgG levels, plasma neutralization on RBD binding, and clinical data were collected from the laboratory and Northwestern EDW database, electronically recorded, and verified by laboratory staff. There were in total $n = 30$ measurable data points for final statistical analyses. To reduce bias resulting from batch effects, four independent replications on RBD-IgG test were performed in the laboratory. One-way ANOVA was performed to compare group means and the replications did not show statistically significant batch or measure errors ($F = 0.01$, $p$-value > 0.9), thus, mean values of the replications were taken for analysis. In addition, a log-linear model (Poisson regression) was fitted to estimate the associations between normalized percentage (%) of RBD binding to cells and independent predictors of interest. It suggested negative associations (see Supplementary Fig. 6b, c) and the adjusted $R^2$ suggests that the combined circulating $ACE2^+$ EV counts + RBD-IgG level explains the relation better than RBD-IgG alone (Adj. $R^2 = 0.623$ $p < 0.0001$). Following linear modeling to determine that combined $ACE2^+$ EVs + RBD-IgG explains the relation better than RBD alone, the relative importance of $ACE2^+$ EVs as compared to anti-RBD IgG was calculated using the Lindeman et al. (1980)[51] formula using the 'relaimpo' package in R (Grömping, 2006)[52]. Metrics were normalized to sum to 100%. The coefficient from this analysis was used to create graphs in Supplementary Fig 6b, c. All statistical analyses were performed by R 4.0.2.

**Animal experiments**. SARS-CoV-2 (nCoV/Washington/1/2020) was provided by the National Biocontainment Laboratory, Galveston, TX. The strain 2019-nCoV/IDF0372/2020 (WT) was supplied by the National Reference Centre for Respiratory Viruses hosted by Institut Pasteur (Paris, France). Animal infection: 6–9 weeks old female and male B6.Cg-Tg(K18-ACE2)2Prlmn/J (K18-hACE2) mice (Jackson Laboratory) were used. All mice were housed in specific pathogen-free facilities, with a regular diet and kept in light from 6 a.m. to 6 p.m. at room temperature (around 22 °C) and humidity 41–42% in the Animal Resources Facilities at BSL-3 facility at University of Chicago Howard T. Ricketts Regional Biocontainment Laboratory. Mice were anesthetized by intraperitoneal injection with ketamine–xylazine (100 mg–20 mg/kg) prior to intranasal administration of EVs and viruses. The suspension of $1 \times 10^4$ PFU of the SARS-CoV-2 virus (10 μL) pre-incubated with EVs (130 μg in 20 μL) for 1 h at 37 °C was administered via drop pipetting into the right nostril of animals. Mice were monitored twice daily to record clinical symptoms and weighed daily for 6 days post-challenge with the virus. Categories in clinical scoring included: Score 0 (pre-inoculation)—animal is bright, alert, active, with a normal fur coat and posture; Score 1 (post-inoculation, pi)—animal is bright, alert, active, normal fur coat and posture, no weight loss; Score 1.5—animal has slightly ruffled fur but is active; weight loss under 2.5%; Score 2 (pi)—animal has ruffled fur, is less active; weight loss under 5%; Score 2.5 (pi)—animal has ruffled fur, is not active but moves when touched, may have hunched posture or difficulty breathing; weight loss 5–10%; Score 3 (pi)—same as score 2.5; weight loss 11–20%; Score 4 (pi)—animal has ruffled fur or is positioned on its side or back, dehydrated, has difficulty breathing; weight loss >20%; Score 5 (pi)—death. At day 6 post-challenge, all animals were euthanized and subjected to necropsy and lung dissection. For each animal, one side of the lungs was homogenized in 2% DMEM to infect Vero-6 cells with serial dilutions for measurement of viral titers (plaque-forming units, pfu), and the other side was fixed with 10% formalin for further histology studies.

**Biodistribution experiment**. B6 mice (Jackson laboratory) (10 weeks old males) were used in the study. Animals were kept in specific pathogen-free facilities with regular diet and regular light/dark cycles, and regular ambient temperature and humidity in the Animal Resources Center at Northwestern University. All animal procedures were complied with the NIH Guidelines for the Care and Use of Laboratory Animals and were approved by the respective Institutional Animal Care and Use Committees. EVs isolated from HEK-ACE2 cells (as described above), were stained with PKH67 (Sigma, PKH67GL-1KT) and PKH67 dilutant control prepared similarly without the presence of EVs. 6–9 weeks old female and male B6 mice (Jackson Laboratory) were deeply anesthetized using isoflurane, and the EV suspension in 25 μL was pipetted into the right nostril of animals. After 24 h, animals were sacrificed, and lungs, brain, heart, liver, kidney, and spleen were isolated, rinsed with PBS. The organs were then imaged and total fluorescence efficiency was quantified using the IVIS imaging system ex vivo imaging was done using LAGO from Spectral Imaging Instruments and images were analyzed using AURA imaging software.

**Histopathological analysis**. Formalin-fixed and paraffin-embedded mouse lungs were processed, sectioned by routine procedures and stained with H&E. Scoring was double-blinded and evaluated by a pathologist based on the percent of total lung surface area involvement (see Supplementary Table 3), following the grading scheme adopted from a previous report[85].

**Statistical analysis**. Microsoft Excel, R 4.0.2, and GraphPad Prism 9.0.2 software were used to perform statistical analyses and calculate the $IC_{50}$. $T$-test or one-way ANOVA followed by Tukey posttest were used where appropriate (such as clear directions of changes). Results are significant if $p < 0.05$. Data are presented as mean ± standard deviation (SD) or standard error of the mean (SEM) (as reported in the figure legends). Parametric analysis was performed unless specified in the figure legends. Measurements were taken from distinct samples in all experiments with biological and/or technical replicates. The cellular components annotation was analyzed by DAVID to compute Fisher's Exact $p$-values.

**Reporting summary**. Further information on research design is available in the Nature Research Reporting Summary linked to this article.

## Data availability
Mass spectroscopy raw data sets have been deposited in the Japan ProteOmeSTandard Repository[83] (https://repository.jpostdb.org/). The accession numbers are PXD029662 for ProteomeXchange[84] and JPST001379 for jPOST. Data that support the findings of this study have been included in the Source Data file. Source data are provided with this paper.

## Code availability
The R program codes used in the data analysis are available in https://github.com/adhoffma/Liu_Lab/tree/main/El-Shennawy%20et%20al%202021.

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

## Acknowledgements

We are thankful to the team of Northwestern COVID-19 Antibody and Cancer Collaborative Group and advisory members, especially Drs. Alfred L. George Jr. (who also edited our manuscript), Judith Varner, Richard D'Aquila, Leonidas C. Platanias, Rex L. Chishom, Alan R. Hauser, Elizabeth M. McNally, and William A. Muller for their scientific input and resourceful support for the project. The work was partially funded by Chicago Biomedical Consortium Accelerator Award A-017 (H.L. and D.F.), the United States National Cancer Institute 1F32CA257345-01 (L.E.), Northwestern University Feinberg School of Medicine Emerging and Re-emerging Pathogens Program (EREPP) (H.L.), Department of Pharmacology Start-up fund (H.L.), the R.H. Lurie Comprehensive Cancer Center Support Grant NIH/NCI CA060553 (S.W.) and Blood Biobank fund (M.I.), and Lyda Hill Philanthropies® (R.K.). We gratefully acknowledge the support from the R.H. Lurie Comprehensive Cancer Center Structural Biology Facility and Flow Core. Flow Cytometry Cell Sorting was performed on a BD FACSAria SORP system, through support of NIH 1S10OD011996-01. The Ametek K3 DDE at Northwestern University for cryo-EM was generously provided by Professor Robert A. Lamb, Ph.D., Sc.D., HHMI investigator. The ACE2-expressing HeLa cells were provided by Dr. Thomas Gallagher of Stritch Medical School, Loyola University. We appreciate the effort of BSL-3 facility at the NIAID-supported University of Chicago Howard T. Ricketts Regional Biochontainment Laboratory for performing the wild-type and variant SARS-CoV-2 live virus study.

## Author contributions

L.E., A.D.H., N.K.D., K.M.M. designed and led the bench experiments, analyzed data, prepared figures, and contributed to writing. P.J.M., D.C., Z.Y., V.L.T., V.N., A.T., C.M., C.J.F., C-F.T., C.O., Y.J., L.L., K.F., X.L., and C.F.R. provided technical support, conducted bench experiments, and analyzed data. J.W. provided HEK-ACE2 cell line and analyzed data. D.B., T.J.H., Y.S., Y.K.C., H.Z., V.S.L., T.S., S.S., Y.L., D.M., G.C.R., A.R.D., M.G.I. provided critical resources and supervised the research project. R.K., D.F., and H.L. designed experiments, analyzed data, wrote the manuscript, and supervised the work.

## Competing interests

MD Anderson Cancer Center and R.K. hold patents in the area of exosome biology (unrelated to the topic of this publication) and are licensed to Codiak Biosciences Inc. MD Anderson Cancer Center and R.K. are stock equity holders in Codiak Biosciences Inc. R.K. is a consultant and a scientific advisor of Codiak Biosciences Inc. Northwestern University and H.L., D.F., L.E., A.D.H., and N.K.D. hold issued and/or provisional (on evACE2) patents in the area of exosome therapeutics. H.L., D.F., and A.D.H. are scientific co-founders in ExoMira Medicine Inc. D.B. and J.W. are co-inventors of patents entitled "Active Low Molecular Weight Variants of Angiotensin-Converting Enzyme 2 (ACE2)", "Active low molecular weight variants of Angiotensin-Converting Enzyme 2 (ACE2) for the treatment of diseases and conditions of the eye" and "Soluble ACE2 Variants and Uses Therefor." D.B. is the founder of Angiotensin Therapeutics Inc. D.B. has received consulting fees from AstraZeneca, Relypsa, and Tricida, all unrelated to this work. J.W. reports scientific advisor capacity for Angiotensin Therapeutics Inc.

## Additional information

[1]Department of Pharmacology, Northwestern University Feinberg School of Medicine, Chicago, IL 60611, USA. [2]Department of Cancer Biology, The University of Texas MD Anderson Cancer Center, Houston, TX 77030, USA. [3]Robert H. Lurie Comprehensive Cancer Center, Northwestern University Feinberg School of Medicine, Chicago, IL 60611, USA. [4]The University of Chicago Howard T. Ricketts Laboratory and Department of Microbiology, Chicago, IL 60637, USA. [5]Division of Health and Biomedical Informatics, Department of Preventive Medicine, Northwestern

University Feinberg School of Medicine, Chicago, IL 60611, USA. [6]Department of Pathology, Northwestern University Feinberg School of Medicine, Chicago, IL 60611, USA. [7]Biological Sciences Division, Pacific Northwest National Laboratory, Richland, WA 99354, USA. [8]Division of Biostatistics, Department of Preventive Medicine, Northwestern University Feinberg School of Medicine, Chicago, IL 60611, USA. [9]Division of Nephrology and Hypertension, Department of Medicine, Northwestern University Feinberg School of Medicine, Chicago, IL 60611, USA. [10]Department of Cell and Developmental Biology, Northwestern University Feinberg School of Medicine, Chicago, IL 60611, USA. [11]Department of Electrical and Computer Engineering, TEES-AgriLife Center for Bioinformatics and Genomic Systems Engineering, Texas A&M University, College Station, TX 77843, USA. [12]Division of Hematology and Oncology, Northwestern University Feinberg School of Medicine, Chicago, IL 60611, USA. [13]Kellogg School of Management, Northwestern University, Evanston, IL 60208, USA. [14]Division of Rheumatology, Department of Medicine, Northwestern University Feinberg School of Medicine, Chicago, IL 60611, USA. [15]Division of Infectious Disease, Department of Medicine, Northwestern University Feinberg School of Medicine, Chicago, IL 60611, USA. [16]Division of Organ Transplantation, Department of Surgery, Northwestern University Feinberg School of Medicine, Chicago, IL 60611, USA. [17]Department of Bioengineering, Rice University, Houston, TX 77005, USA. [18]Department of Molecular and Cellular Biology, Baylor College of Medicine, Houston, TX 77030, USA. [19]These authors contributed equally: Lamiaa El-Shennawy, Andrew D. Hoffmann, Nurmaa K. Dashzeveg, Kathleen M. McAndrews. [20]These authors jointly supervised this work: Raghu Kalluri, Deyu Fang, Huiping Liu. ✉email: RKalluri@mdanderson.org; fangd@northwestern.edu; huiping.liu@northwestern.edu

