## [Peer Review File · Nature Communications]

Circulating ACE2-expressing Extracellular Vesicles as A Decoy Therapy to Block Broad Strains of SARS-CoV-2Editorial Note: This manuscript has been previously reviewed at another journal that is not operating a transparent peer review scheme. This document only contains reviewer comments and rebuttal letters for versions considered at Nature Communications.

Reviewers' Comments:

Reviewer #1:

Remarks to the Author:

We have previously reviewed this article for Nature, and we noticed that the authors have included several new experiments and answered our previous requests in this new revised version. The article is consequently more convincing and brings more novel aspects.

There are still some things that should be amended and corrected before this paper is suitable for publication.

- 1) As currently presented, the authors do not acknowledge previous work that had shown relevant results for their study: the authors MUST quote the previously published articles that had already shown that ACE2 is released on EVs from cells expressing ACE2 naturally, or after overexpression (including in HEK cells, as done here), and that these ACE2+ EVs bound the spike and/or decreased infection by a spike-pseudotyped lentivirus: Zhang et al, *Gastroenterology* 2021 # 33022277, and Coccozza et al, *J Extracell Vesicles* 2020 # 33391636.
- 2) Another interesting paper showing the presence of circulating ACE2-EVs in patients should be also discussed, in light of the authors results: Krishnamachary, B., et al. (2021). "Extracellular vesicle-mediated endothelial apoptosis and EV-associated proteins correlate with COVID-19 disease severity." *J Extracell Vesicles* 10(9): e12117.
- 3) The authors now use properly the term "EVs" rather than exosomes, except in line 141, referring to the gradient of extfig2e: in such a gradient, MVB-derived exosomes and other EVs are floating at similar densities, and CD81 is not specific of exosomes.
- 4) The authors have done several experiments to now convincingly show that there is no soluble/cleaved ACE2 in the EVs isolated from ACE2-expressing cells, however, the blots shown in fig 1f and ext 1e, 2c, 2e must show a larger part including proteins of sizes below 80kDa (where the expected cleaved ACE2 should be).
- 5) In the flow vesiclotometry analysis, the authors wrongly mention an enrichment of ACE2 in CD63+ EVs, when they show that only about 30% of CD63+ EVs are ACE2+, and the plot of ext3c suggests that in fact the number of ACE2+ EVs is higher than the number of CD63+ EVs, and of CD81+ EVs (these ones show a weird cloud of SALS-low CD81+ events: are these EVs?). This suggests that ACE2 is in fact present on some CD63+ EVs, some CD81+ EVs, and probably other EVs bearing neither one. In addition, the M&M does not indicate what antibody and conditions are used for staining CD63 in flow vesiclotometry.

Other comments:

New exp fig3d shows inhibition of infection by real virus using EVs from plasma of patients when ACE2 detected. Is it then consistent with stronger detection of ACE2-EVs in plasma of acute phase patients, should these patients not be protected when they have more ACE2-EVs??

The new data on hACE2 mouse models are useful, however, infection with SARS-CoV2 (fig4b) shows only 2/5 mice containing high virus load after injection of control EVs, thus this experiment is not very strong! Fig4a with cohorts of 20 mice is much more convincing.

Data analyzing effect of ACE2-EVs on infection by the new SARS-CoV-2 variants are an interesting novel feature.

Reviewer #2:

Remarks to the Author:

The study by El-Shennawy et al reports ACE2-positive EVs (evACE2) in the plasma of both acute and convalescent COVID-19 patients. Furthermore, they show that evACE2 are able to neutralize SARS-CoV-2 infection by competing with cellular ACE2. In vitro experiments demonstrate the potency of evACE2 to block the binding of the viral spike (S) protein RBD to ACE2+ cells and reduce infectivity significantly. In vivo data using hACE2 mice indicate that evACE2 protect from SARS-CoV-2-induced lung injury and mortality. They inhibit the infection of SARS-CoV-2 wildtype strain as well as of SARS-CoV-2-variants (a, b and d) with equal potency.

The additional data greatly improved the former manuscript, and all concerns raised have been adequately addressed. The results presented suggest that the release of evACE2 by infected cells may function as a defense mechanism against SARS-CoV-2 infection. The data confirm results published by other groups (Zhang et al, *Gastroenterology*, 2021, PMID: 33022277; Cocozza et al, *JEV*, 2020, PMID: 33391636). But the study also includes information regarding the blocking efficiency of evACE2 and additional data on clinical relevance. Furthermore, El-Shennawy and colleagues now provide additional in vitro data. While Wu et al. have previously shown that intranasal pretreatment with ACE+ EV can suppress S-pseudovirus entry into the mucosal epithelium (Wu et al, *Acta Pharm Sin B*, 2021, #34522576), the present study further demonstrates significantly reduced lung epithelial infection and, as a result, significantly reduced lung inflammation in the mice infected with SARS-CoV-2. However, it is yet unclear whether evACE2 is indeed effective as a therapeutic agent against SARS-CoV-2 infection. The experimental design of the in vivo experiments does not allow any conclusions to be drawn regarding a therapeutic benefit, as viruses were incubated with evACE2 prior to intranasal application and thus were already neutralized ex vivo. To address the question of a therapeutic benefit, evACE2 treatment would need to be performed in infected mice.

Minor comments:

1. Figure 1: It would be interesting to know, if the documented co-expression of CD81 and ACE2 on cell-line derived evACE2 (30-60%) is also observed on plasma EV of COVID-19 patients.
2. Extended Figure 1e: The depicted immunoblot indicates that EV release is triggered in general. Lack of TSG101 in non-infected cells indicates that there are hardly any endosomal pathway-derived EVs present.
3. Extended Figure 3f: How was the amount of ACE2 exactly estimated? What is the difference to Extended Fig. 3d? Please clarify.
4. Line 131 - Syntenin-1 immunoblot is not depicted anywhere
5. Line 133 - refers to non-existing Extended Figure 1f

Reviewer #3:

None

Thank you for your consideration of our manuscript NCOMMS-21-37264-A, newly entitled “Circulating ACE2-expressing Extracellular Vesicles as A Decoy Therapy to Block Broad Strains of SARS-CoV-2. We appreciate your remarks and comments, and we are addressing them point by point.

Reviewer #1 (Remarks to the Author):

We have previously reviewed this article for Nature, and we noticed that the authors have included several new experiments and answered our previous requests in this new revised version. The article is consequently more convincing and brings more novel aspects.

There are still some things that should be amended and corrected before this paper is suitable for publication.

1) As currently presented, the authors do not acknowledge previous work that had shown relevant results for their study: the authors MUST quote the previously published articles that had already shown that ACE2 is released on EVs from cells expressing ACE2 naturally, or after overexpression (including in HEK cells, as done here), and that these ACE2+ EVs bound the spike and/or decreased infection by a spike-pseudotyped lentivirus: Zhang et al, Gastroenterology 2021 # 33022277, and Coccozza et al, J Extracell Vesicles 2020 # 33391636.

We thank the reviewer for reminding us of this important detail. We cited the references suggested by the reviewer in our article (Ref 58 and 59).

2) Another interesting paper showing the presence of circulating ACE2-EVs in patients should be also discussed, in light of the authors results: Krishnamachary, B., et al. (2021). "Extracellular vesicle-mediated endothelial apoptosis and EV-associated proteins correlate with COVID-19 disease severity." J Extracell Vesicles 10(9): e12117.

We agree with the reviewer that the suggested paper should be discussed in light of our findings. We included the suggested reference in our discussion (Ref 70).

3) The authors now use properly the term “EVs” rather than exosomes, except in line 141, referring to the gradient of extfig2e: in such a gradient, MVB-derived exosomes and other EVs are floating at similar densities, and CD81 is not specific of exosomes.

We thank the reviewer for this remark. In line 141 we have replaced “exosomes” with “small EV” to avoid confusion:

“The full-length ACE2 was almost exclusively detected in small EV fractions co-expressing CD81, with minimal or no detectable ACE2 in the non-vesicular fractions that express the putative exomere marker HSP90 (Supplementary Fig. 2d-e). These results indicate that evACE2 is enriched in small EVs with minimal detection in presumed exomeres.”

We would like to clarify that we used density gradient fractionation to separate small extracellular vesicles (sEVs) from non-vesicular components (NVs). Gradient-purified sEVs are highly distinct from NVs in terms

of proteomic profiling as described by Jeppesen et al, Cell, 2019. Common exosomal markers such as CD81, are relatively overexpressed in gradient-purified sEVs. On the other hand, distinct association of histones, such as HSP90, with NV fractions has been reported.

4) The authors have done several experiments to now convincingly show that there is no soluble/cleaved ACE2 in the EVs isolated from ACE2-expressing cells, however, the blots shown in fig 1f and ext 1e, 2c, 2e must show a larger part including proteins of sizes below 80kDa (where the expected cleaved ACE2 should be).

We thank the reviewer for the remark. In the source data of full western blots (from 20-250 kD) for figure 1f, supplementary Fig 1e and 2c (see below), little to undetectable proteins were shown below 80kD. In addition, the soluble ACE2 control (right panel) shows a molecular weight below and distinct from the full length ACE2 which is estimated 110-120 kD, possibly resulted from post-translational modifications, such as glycosylation. In Supplementary figures 2c, the rhACE2 bands appear at higher molecular weight, partially due to different gels, different ladder controls, probed with a different anti-ACE2 antibody (4-12% gels, Cell signaling 4355s antibody) compared to Figures 1f and Supplementary figure 1e (4-20% gels, R&D AF933).

Figure 1f

Supplementary Figure 1e

Suppl Figure 2c

5) In the flow vesicometry analysis, the authors wrongly mention an enrichment of ACE2 in CD63+ EVs, when they show that only about 30% of CD63+ EVs are ACE2+, and the plot of ext3c suggests that in fact the number of ACE2+ EVs is higher than the number of CD63+ EVs, and of CD81+ EVs (these ones show a weird cloud of SALS-low CD81+ events: are these EVs?). This suggests that ACE2 is in fact present on some CD63+ EVs, some CD81+ EVs, and probably other EVs bearing neither one. In addition, the M&M does not indicate what antibody and conditions are used for staining CD63 in flow vesicometry.

- We thank the reviewer for drawing our attention to this point. We removed the word “enrichment” from the text and the sentence is now: “Double staining MFV analyses detected ACE2 in CD81+ EVs (31.9 - 62.5%) or CD63+ EVs (33.2 - 47.8%) for HEK-ev1 and HeLa-ev2 (Fig. 1h, i)”.
- We agree with the reviewer that our results suggest that ACE2 is present in some- and not all- CD63+ or CD81+ EVs and other EVs might not be bearing neither one. In fact, we are detecting markers within the sensitivity level/limits of the Apogee. Moreover, extracellular vesicles are heterogeneous and not all vesicles express those same proteins.
- The weird cloud of SALS-low CD81+ events are expected to be a population of EVs in small size.
- Our updated manuscript is now including details about the antibodies used for flow vesicometry in the M&M section.

Other comments:

New exp fig3d shows inhibition of infection by real virus using EVs from plasma of patients when ACE2 detected. Is it then consistent with stronger detection of ACE2-EVs in plasma of acute phase patients, should these patients not be protected when they have more ACE2-EVs??

We agree with the reviewer that detection of high ACE2-EVs levels in plasma of acute patients would suggest that these ACE2-EVs might protect from new viral infection. However, the inhibition by ACE2-EVs requires high dosage and appropriate timing that might not be sufficient or optimal when the viral load is high along with over-reacting immune responses in those patients, like the high titers of neutralizing antibodies in the patient plasma. In addition, severe COVID-19 is most associated with overreacting inflammatory response that may not be inhibited by evACE2.

The new data on hACE2 mouse models are useful, however, infection with SARS-CoV2 (fig4b) shows only 2/5 mice containing high virus load after injection of control EVs, thus this experiment is not very strong! Fig4a with cohorts of 20 mice is much more convincing.

We thank the reviewer for this important remark. We agree that results in figure 4b shows only 2/5 mice containing high virus load after of control EVs. Our explanation to this observation is that for each animal, one of the two lungs only was used to determine the viral load, while the other lung of the same animal was used for H&E staining. Thus, it is possible that the viral count in one lung is not as high as the other at the time animals were sacrificed, dependent on the distribution of the inhaled viruses.

Data analyzing effect of ACE2-EVs on infection by the new SARS-CoV-2 variants are an interesting novel feature.

We thank the reviewer for this comment. Our results support a broad-spectrum antiviral mechanism of evACE2 for therapeutic development to block the infection of existing and future corona viruses that use the ACE2 receptor.

Reviewer #2 (Remarks to the Author):

The study by El-Shennawy et al reports ACE2-positive EVs (evACE2) in the plasma of both acute and convalescent COVID-19 patients. Furthermore, they show that evACE2 are able to neutralize SARS-CoV-2 infection by competing with cellular ACE2. In vitro experiments demonstrate the potency of evACE2 to block the binding of the viral spike (S) protein RBD to ACE2+ cells and reduce infectivity significantly. In vivo data using hACE2 mice indicate that evACE2 protect from SARS-CoV-2-induced lung injury and mortality. They inhibit the infection of SARS-CoV-2 wildtype strain as well as of SARS-CoV-2-variants (a, b and d) with equal potency.

The additional data greatly improved the former manuscript, and all concerns raised have been adequately addressed. The results presented suggest that the release of evACE2 by infected cells may function as a defense mechanism against SARS-CoV-2 infection. The data confirm results published by other groups (Zhang et al, Gastroenterology, 2021, PMID: 33022277; Cocozza et al, JEV, 2020, PMID: 33391636). But the study also includes information regarding the blocking efficiency of evACE2 and additional data on clinical relevance. Furthermore, El-Shennawy and colleagues now provide additional in vitro data. While Wu et al. have previously shown that intranasal pretreatment with ACE+ EV can suppress S-pseudovirus entry into the mucosal epithelium (Wu et al, Acta Pharm Sin B, 2021, #34522576), the present study further demonstrates significantly reduced lung epithelial infection and, as a result, significantly reduced lung inflammation in the mice infected with SARS-CoV-2.

However, it is yet unclear whether evACE2 is indeed effective as a therapeutic agent against SARS-CoV-2 infection. The experimental design of the in vivo experiments does not allow any conclusions to be drawn regarding a therapeutic benefit, as viruses were incubated with evACE2 prior to intranasal application and thus were already neutralized ex vivo. To address the question of a therapeutic benefit, evACE2 treatment would need to be performed in infected mice.

We thank the reviewer for this important remark. We have toned down that our data provide proof-of-concept evidence for “therapeutic development” that the live virus is neutralized by evACE2, and that evACE2 protects the hACE2 transgenic mice from SARS-CoV-2-induced lung injury and mortality, as shown in the last part of Abstract:

Consistently, evACE2 protects the hACE2 transgenic mice from SARS-CoV-2-induced lung injury and mortality. Furthermore, evACE2 inhibits the infection of SARS-CoV-2 variants (α , β , and δ) with equal or higher potency than for the wildtype strain, supporting a broad-spectrum antiviral mechanism of evACE2 for therapeutic development to block the infection of existing and future corona viruses that use the ACE2 receptor.

With the results of our current animal experiment, we would propose to test the therapeutic potential of evACE2 to block the infection of corona viruses in the future animal studies, which are beyond the scope of this study but can evaluate the role of evACE2 in treating infected mice and assessing the therapeutic benefit.

Minor comments:

1. Figure 1: It would be interesting to know, if the documented co-expression of CD81 and ACE2 on cell-line derived evACE2 (30-60%) is also observed on plasma EV of COVID-19 patients.

We agree with the reviewer that it is important to show the co-expression of an EV marker and ACE2 on plasma EV of COVID-19 patients. We would like to draw the attention of the reviewer to figure 1c which demonstrates the expression of ACE2 within CD63+ EVs in patients' plasma.

Fig 1c:

2. Extended Figure 1e: The depicted immunoblot indicates that EV release is triggered in general. Lack of TSG101 in non-infected cells indicates that there are hardly any endosomal pathway-derived EVs present.

We agree with the reviewer with this observation. We therefore only concluded that "...implying a possibility that either the virial pathogens, or their associated pathogenesis, induce the generation of evACE2" (in discussion) without specifying the endosomal pathway.

Nevertheless, recent literature reported that viral infection enhances the release of endosomal pathway-derived EVs, such as modulating the production and content of EVs in the host cells (Raab-Traub and Dittmer, Nature Reviews microbiology, 2017, Ref 71). Moreover, an interesting article has recently shown that that ORF3a of SARS-CoV-2, but not SARS-CoV, promotes lysosomal exocytosis (Chen et al., Dev Cell, 2021, Ref 72). We have included these two papers in discussion.

Nevertheless, viral infection may modulate the production and content of EVs⁷¹ and SARS-CoV-2 has been shown to promote lysosomal exocytosis⁷².

3. Extended Figure 3f: How was the amount of ACE2 exactly estimated? What is the difference to Extended Fig. 3d? Please clarify.

We isolated EVs from ACE2 overexpressing cells and measured EV-protein in PBS using nanodrop. EVs were lysed and centrifuged. Then we used the samples to estimate the amount of ACE2 in EVs using two different methods:

First: We used Human ACE-2 ELISA kit (RayBiotech, ELH-ACE2-1) where the antibody pair detects extracellular domain of Human ACE-2. We determined the ACE2 content in different batches of isolated EVs as well as the commercial rhACE2 we used for our experiments. The results are shown in extended/supplementary figure 3d for EVs isolated from HEK-ACE2 cells (ev1) and HeLa-ACE2 cells (ev2).

Second: We used Western Blotting, where the lysed EVs were denatured and loaded to SDS-PAGE then transferred to nitrocellulose membranes that were incubated O/N with the ACE2 primary antibody (R&D systems, AF933). We loaded different concentrations (serial dilutions) of the commercial rhACE2 as well. Using densitometry quantification, we created a standard curve for rhACE2 and used it to estimate ACE2 content in the loaded EVs (supplementary figure 3d). Supplementary figure 3f represents the results of densitometry quantification.

Finally: We detected a similar range of ACE2 content in the isolated EVs, including 0.1-0.2 ng ACE2 per μ g EV protein of HEK-ev1 and HeLa-ev2 using both ELISA and Western Blotting.

4. Line 131 - Syntenin-1 immunoblot is not depicted anywhere

We apologize if we were not clear. Syntenin-1 is depicted in supplementary figure 2c where blots were probed with Syntenin-1.

5. Line 133 - refers to non-existing Extended Figure 1f

In line 133: “marker *GRP94* (Fig. 1f, Supplementary Fig. 2b-e). We also confirmed that *evACE2 purification*”, we refer to Figure 1f and not to extended/supplementary figure 1f.